# Anomalous Hall effect from inter-superlattice scattering in a noncollinear antiferromagnet

Lilia S. Xie [1,9] ✉, Shannon S. Fender [1], Cameron Mollazadeh[1], Wuzhang Fang [2], Matthias D. Frontzek [3], Samra Husremović [1], Kejun Li [2,4], Isaac M. Craig [1], Berit H. Goodge [1,5], Matthew P. Erodici[1], Oscar Gonzalez[1], Jonathan D. Denlinger [6], Yuan Ping [2] & D. Kwabena Bediako [1,7,8] ✉

Superlattice formation dictates the physical properties of many materials, including the nature of the ground state in magnetic materials. Chemical composition is commonly considered to be the primary determinant of superlattice identity, especially in intercalation compounds. Nevertheless, in this work, we find that kinetic control of superlattice growth leads to the coexistence of disparate crystallographic domains within a compositionally perfect single crystal. We demonstrate that $Cr_{1/4}TaS_2$ is a noncollinear anti-ferromagnet in which scattering between majority and minority superlattice domains engenders complex magnetotransport below the Néel temperature, including an anomalous Hall effect. We characterize the magnetic phases in different domains, image their nanoscale morphology, and propose a mechanism for nucleation and growth using a suite of experimental probes coupled with first-principles calculations and symmetry analysis. These results provide a blueprint for the deliberate engineering of macroscopic transport responses via microscopic tuning of magnetic exchange interactions in superlattice domains.

Superlattices impart emergent properties to many materials, including metal alloys[1], semiconductors[2], and ceramics[3]. The periodicity and symmetry of the long-range order determine the underlying many-body physics, with especially profound implications for quantum materials. Moiré superlattices, for example, host a plethora of emergent electronic phenomena, such as unconventional superconductivity[4], correlated insulating states[5,6], and the quantum anomalous Hall effect[7,8]. In magnetic materials, superlattices of the spin-bearing centers can dictate the distances between them,

strongly influencing the microscopic magnetic exchange interactions and hence the ground state[9–11].

The link between superlattice identity and magnetic ordering is especially apparent in transition metal dichalcogenides (TMDs) intercalated with first-row transition metals[12–14]. In this family of materials, varying the host lattice, intercalant, and stoichiometry allows access to many types of magnetism, including hard ferromagnetism[15–17], non-collinear antiferromagnetism[18–20], and spin glass phases[21,22]. Although off-stoichiometry and structural disorder are known to modify their

[1]Department of Chemistry, University of California, Berkeley, CA 94720, USA. [2]Department of Materials Science and Engineering, University of Wisconsin, Madison, WI 53706, USA. [3]Neutron Scattering Division, Oak Ridge National Laboratory (ORNL), Oak Ridge, Tennessee 37831, USA. [4]Department of Physics, University of California, Santa Cruz, CA 95064, USA. [5]Max-Planck-Institute for Chemical Physics of Solids, Nöthnitzer Str. 40, 01187 Dresden, Germany. [6]Advanced Light Source, Lawrence Berkeley National Laboratory, Berkeley, CA 94720, USA. [7]Chemical Sciences Division, Lawrence Berkeley National Laboratory, Berkeley, CA 94720, USA. [8]Kavli Energy NanoScience Institute, Berkeley, CA 94720, USA. [9]Present address: Department of Chemistry and the Princeton Materials Institute, Princeton University, Princeton, NJ 08544, USA. ✉e-mail: liliaxie@princeton.edu; bediako@berkeley.edu

magnetotransport properties[23–25], composition is generally treated as a proxy for the periodicity of the intercalant superlattice. Assuming a formula of $T_xMCh_2$, where $T$ is a first-row transition metal, $M$ is Nb or Ta, and $Ch$ is S or Se, it is typically assumed that $x = 1/4$ results in a $2 \times 2$ superlattice, $x = 1/3$ results in a $(\sqrt{3} \times \sqrt{3})R30°$ superlattice, and intermediate compositions result in defective, disordered, or mixed superlattices[24,26–30]. Nevertheless, since the magnetism of these materials is highly sensitive to the intercalant superlattice[26,31,32], precise control over atomic-scale ordering is necessary to realize transport signatures associated with specific magnetic phases, including noncollinear textures and altermagnetism[32–36].

In this work, we synthesize the new material $Cr_{1/4}TaS_2$ and find that it is a bulk noncollinear antiferromagnet with a well-ordered $2 \times 2$ superlattice of Cr. Below the Néel temperature ($T_N$) of 145 K, it exhibits an anomalous Hall response, which is inconsistent with the symmetry of the magnetic structure. Detailed characterization of the nanoscale atomic structure reveals minority domains with different superlattice ordering, even in high-quality, stoichiometric single crystals. We propose a mechanism for kinetically arrested growth of disparate superlattice domains and show that the complex magnetotransport originates from scattering between domains. The results show how variation of local ordering can be leveraged to tune magnetotransport in an antiferromagnet, with implications for designing transport responses via deliberate superlattice engineering without changes in chemical composition.

## Results

### Synthesis and crystallographic and magnetic structure

We initially targeted the compound $Cr_{1/4}TaS_2$ with the hypothesis that $Cr^{3+}$ ($S = 3/2$) is a promising intercalant for targeting a noncollinear ground state: it has a largely quenched orbital moment[37] and strong easy-plane anisotropy[38], which can lead to predictable interplay between second-order perturbations and Heisenberg exchange. For example, in $Cr_{1/3}NbS_2$ and $Cr_{1/3}TaS_2$, the Dzyaloshinskii–Moriya interaction arising from broken inversion symmetry competes with ferromagnetic coupling, resulting in chiral helimagnetism[39–41]. We thus surmised that putting $Cr^{3+}$ on a geometrically frustrated triangular lattice with antiferromagnetic (AFM) coupling would favor a noncollinear ground state to relieve the frustration (Fig. 1a), potentially yielding an anomalous Hall response[18–20,42–44].

Single crystals of $Cr_{1/4}TaS_2$ were grown from the constituent elements using chemical vapor transport with iodine as a transport agent. We found that Cr crystallographic disorder was minimized in a two-zone furnace with the hot zone at 1100 °C and the cold (growth) zone at 1000 °C, and a cooling rate of 20 °C/h. The structure, as determined by single-crystal X-ray diffraction (SCXRD), consists of $2H$-$TaS_2$ layers with 1/4 of the pseudo-octahedral sites in the van der Waals gap occupied by Cr, forming a $2 \times 2$ superlattice (Fig. 1b; more details of the refinement in Tables S1 and S2). Within the sensitivity limits of a laboratory diffractometer, we detect no evidence of Cr deficiency on the $2a$ site, electron density on the $6g$ site, or additional reflections corresponding to the $(\sqrt{3} \times \sqrt{3})R30°$ superlattice (Fig. S1). Unless otherwise indicated, all experiments were carried out on crystals from two batches with perfect (within experimental error) $2 \times 2$ Cr stoichiometries of $x = 0.252(3)$ as determined by energy dispersive X-ray spectroscopy (Fig. S2), and a fully occupied $2 \times 2$ superlattice as determined by SCXRD (see the Methods for more details on crystal growth and compositional characterization).

Heat capacity and neutron diffraction measurements establish that $Cr_{1/4}TaS_2$ is a bulk antiferromagnet with $T_N = 145$ K. A single $\lambda$ anomaly in the heat capacity ($C_p$) is observed at 145 K (Fig. 1c and inset), indicating a bulk phase transition at this temperature. Using single-crystal neutron diffraction in the ($HHL$) scattering plane, we observe satellite peaks below 150 K consistent with the magnetic propagation

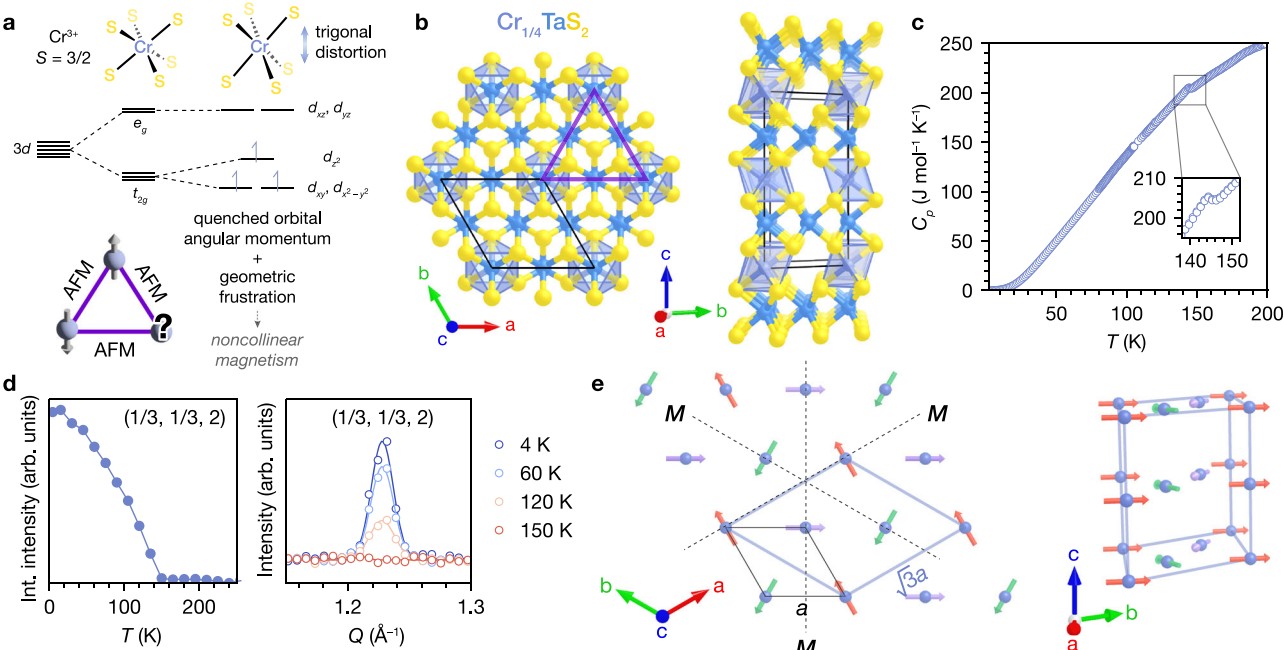

**Fig. 1 | $Cr_{1/4}TaS_2$ is a noncollinear antiferromagnet. a** Local $d$-orbital splitting for $Cr^{3+}$, and design principle for noncollinear magnetism through geometrically frustrated antiferromagnetic (AFM) interactions. **b** Structure of $Cr_{1/4}TaS_2$ from single-crystal X-ray diffraction. The unit cell ($2 \times 2 \times 1$ relative to $2H$-$TaS_2$) is indicated in black and the triangular lattice is emphasized in purple. **c** Heat capacity ($C_p$) vs. $T$ normalized to the formula $CrTa_4S_8$. **d** Single-crystal neutron diffraction data: Integrated intensities of the (1/3, 1/3, 2) peak vs. $T$, and the (1/3, 1/3, 2) peak at different temperatures (solid lines indicate Gaussian fits), associated with the propagation vector $\mathbf{k} = (1/3, 1/3, 0)$. **e** 120° AFM structure as obtained from neutron diffraction ($\Gamma_6$ representation). The $(\sqrt{3} \times \sqrt{3})R30° \times 1$ magnetic unit cell (relative to the $2 \times 2 \times 1$ Cr superlattice) is shown in light blue, and the nuclear unit cell is shown in black. The magnetic structure is invariant with respect to the three mirror planes perpendicular to the magnetic moments (dotted black lines).

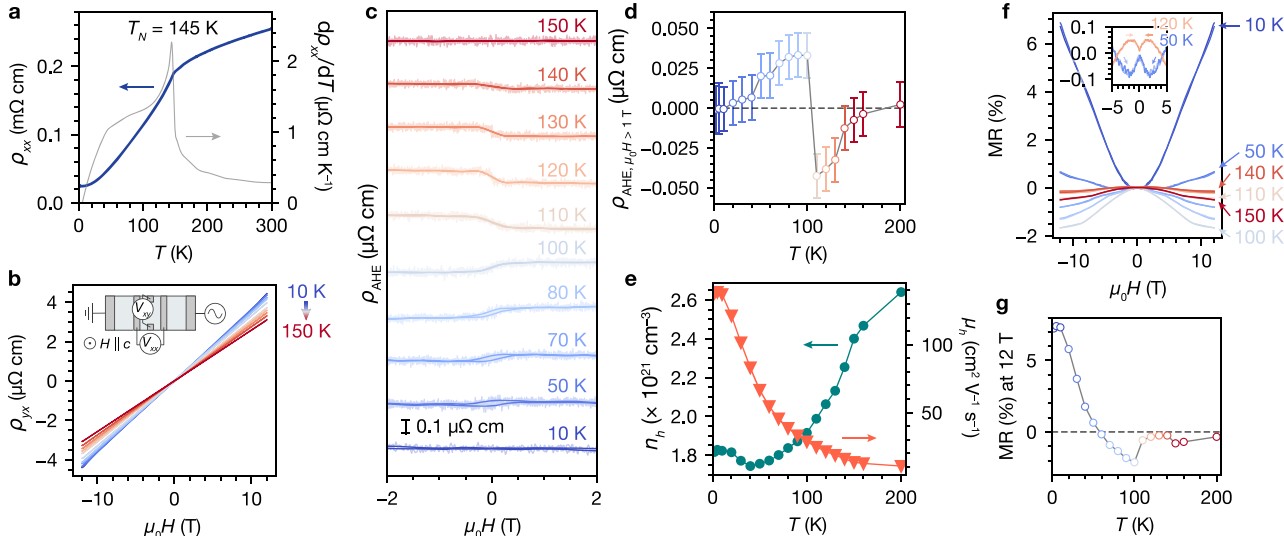

**Fig. 2 | Magnetotransport properties of $Cr_{1/4}TaS_2$. a** Longitudinal resistivity ($\rho_{xx}$) and $d\rho_{xx}/dT$ vs. $T$. **b** Hall resistivity ($\rho_{yx}$) vs. $\mu_0H$, and schematic of the measurement configuration. Colors indicate temperatures as labeled in (**c**). **c** Anomalous Hall resistivity ($\rho_{AHE}$) vs. $\mu_0H$ at different temperatures. Translucent lines are the raw data; opaque lines are denoised. **d** The average high-field values of $\rho_{AHE}$ ($\mu_0H > 1$ T) vs. $T$. Error bars are the standard deviation of $\rho_{AHE}$ between 2 and 5 T. **e** Charge carrier concentration ($n_h$) and carrier mobility ($\mu_h$) vs. $T$, as derived from the ordinary Hall component of $\rho_{yx}$. **f** Magnetoresistance (MR) vs. $\mu_0H$ at different temperatures. Colors indicate temperatures as labeled in (**c**). **g** MR at 12 T vs. $T$.

vector $\mathbf{k} = (1/3, 1/3, 0)$, corresponding to the onset of AFM ordering (Fig. 1d). (Due to crystal size requirements, we measured a slightly Cr-deficient crystal with $x = 0.226(6)$.) Representation analysis of the observed $\mathbf{k}$ using SARA$h$[45] indicates that the magnetic representation $\Gamma_{mag}$ can be decomposed into six irreducible representations ($\Gamma_{mag} = \Gamma_1 + \Gamma_2 + \Gamma_3 + \Gamma_4 + 2\Gamma_5 + 2\Gamma_6$), of which $\Gamma_5$ and $\Gamma_6$ are consistent with the easy-plane anisotropy. The only stable refinement of data collected at 1.5 K is obtained with $\Gamma_6$, which corresponds to an in-plane cycloidal structure with parallel orientation of moments along $c$. When an equal moment constraint is applied, the value refines to $2.07(8)$ $\mu_B$/Cr, with the spins oriented along the $\langle 100 \rangle$ directions, i.e. a $120°$ AFM structure (Fig. 1e). The lower-than-expected moments (compared to the theoretical $3\mu_B/Cr^{3+}$) could be a consequence of itinerant magnetism, which has been observed in related materials[35,36,46]. Additional details of the refinement are available in the Supplementary Information (Tables S3–S5). Altogether, the heat capacity and neutron diffraction data unambiguously point to a single bulk AFM transition in $Cr_{1/4}TaS_2$ at 145 K.

## Magnetotransport

Electrical transport measurements on bulk single crystals of $Cr_{1/4}TaS_2$ reveal metallic behavior from the longitudinal resistivity ($\rho_{xx}$) (Fig. 2a). A kink at $T_N = 145$ K is consistent with reduced carrier scattering upon bulk AFM ordering. The residual resistivity ratio, $RRR = \rho_{300K}/\rho_{2.5K}$, is about 10, indicating good crystal quality. The Hall resistivity ($\rho_{yx}$) is dominated by the ordinary Hall effect (Fig. 2b). An anomalous Hall contribution, evidenced by jumps in $\rho_{yx}$ centered at zero field, is present below $T_N$ (Fig. 2c, d). We fit $\rho_{yx}$ for fields < 5 T to a single-band model comprising an ordinary Hall component and a small component from an anomalous Hall effect (AHE), $\rho_{yx} = \frac{1}{ne}\mu_0H + \rho_{AHE}$. From the ordinary Hall component, we extract carrier concentrations ($n_h$) for the dominant hole carriers on the order of $10^{21}$ cm$^{-3}$, and carrier mobilities ($\mu_h$) up to 138 cm$^2$ V$^{-1}$ s$^{-1}$ at 2.5 K (Fig. 2e), reasonable values for an electron-doped intercalation compound of $2H$-TaS$_2$[16]. The faster decrease in $n_h$ below 150 K may correspond to Fermi surface reconstruction as a result of AFM ordering[47].

Several interesting features are observed in the temperature dependence of $\rho_{AHE}$ (Fig. 2c, d). Below 40 K, $\rho_{AHE}$ is negligible. Between

40–100 K, $\rho_{AHE}$ is positive in sign and exhibits a small coercive field of <1 T up to 80 K. Above $\mu_0H > 1$ T, the $\rho_{AHE}$ values are approximately constant; we plot the average values at each temperature in Fig. 2d. At 100 K and below, these average high-field values of $\rho_{AHE}$ increase with increasing $T$. At 110 K, the circulation of $\rho_{AHE}$ changes sign while retaining a similar magnitude. $\rho_{AHE}$ then approaches zero with increasing temperature and vanishes above $T_N$. We calculate the anomalous Hall conductivity, $\sigma_{AHE}$, by first calculating the total transverse conductivity, $\sigma_{xy} = \frac{\rho_{yx}}{\rho_{yx}^2 + \rho_{xx}^2}$, and then subtracting the ordinary Hall component. The maximum $\sigma_{AHE}$ is about 8 $\Omega^{-1}$ cm$^{-1}$ at 50 K.

The magnetoresistance (MR), defined as $\Delta\rho_{xx}(T, H)/\rho_{xx}(T, 0)$, also exhibits a complex temperature dependence (Fig. 2f, g). At 10 K and below, the MR is positive and approximately quadratic with respect to field. Above 10 K, the MR decreases quickly with increasing $T$. The high-field MR becomes negative at 60 K and increases gradually in magnitude until 100 K. There is a significant decrease in the magnitude of the negative MR between 100 and 110 K. At intermediate temperatures in the positive and negative MR regimes, cusps are observed around zero field (Fig. 2f inset). At 150 K and above, i.e. above $T_N$, the MR becomes uniformly quadratic in field and remains negative.

The fact that a non-zero $\rho_{AHE}$ and complex temperature-dependent MR are observed only below $T_N$ indicates that these magnetotransport phenomena are tied to the bulk AFM order. The intrinsic anomalous Hall conductivity (AHC) can be evaluated from the integral of Berry curvature over the Brillouin zone within linear response theory. However, symmetry analysis of the magnetic structure shows that it is invariant with respect to the mirror planes perpendicular to the magnetic moments, as indicated by the dashed lines in Fig. 1e. The Berry curvature is odd under these mirror symmetry transformations. Therefore, the intrinsic AHC sums to zero across the Brillouin zone[48]. More details of the symmetry analysis and the calculated AHC are presented in the Supplementary Information (Equations S1–S5 and Fig. S3). Hence, we sought to investigate mechanisms for an extrinsic, scattering-mediated AHE that could also explain the rich MR behavior.

## Local superlattice ordering

Considering the nature of $Cr_{1/4}TaS_2$ as an intercalation compound, we used more sensitive probes of local symmetry to study possible

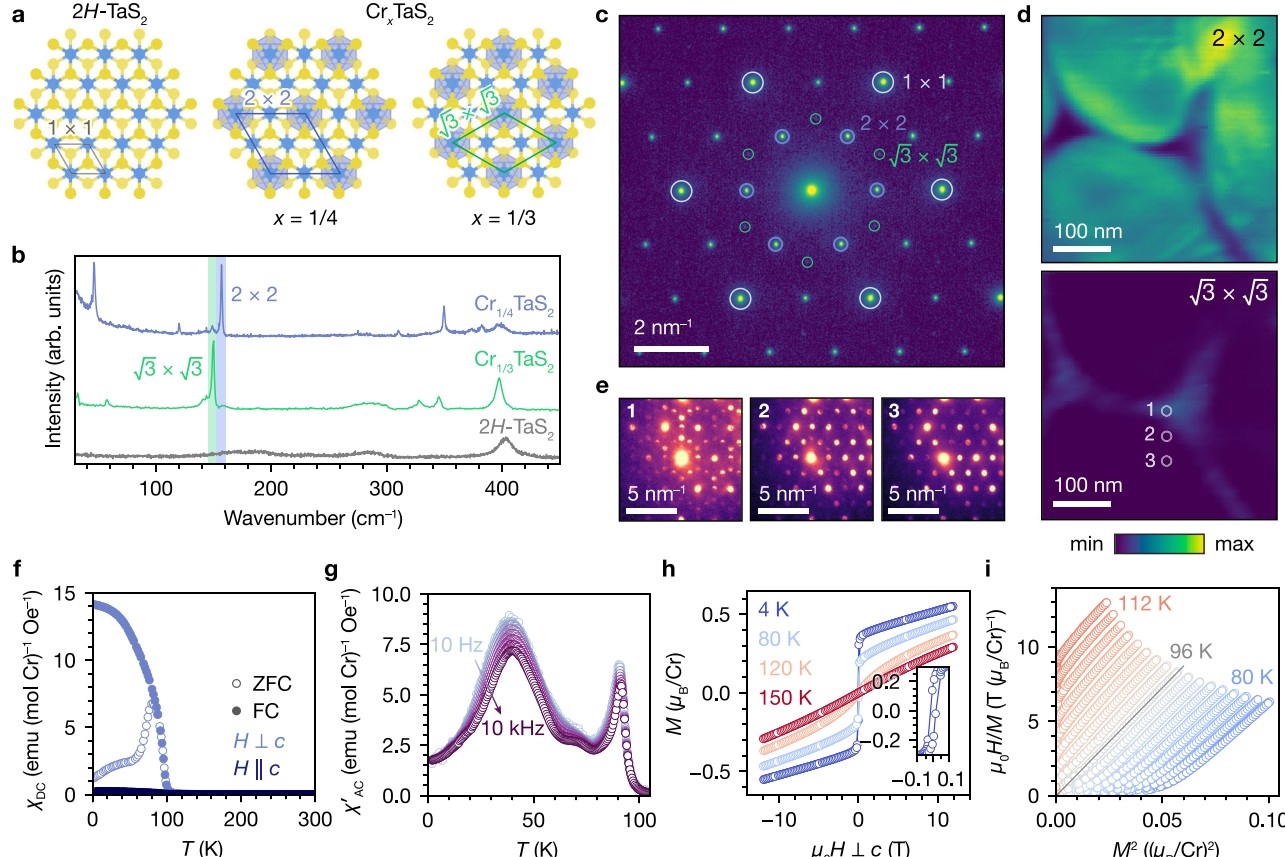

**Fig. 3 | Evidence for $\sqrt{3} \times \sqrt{3}$ domains in $Cr_{1/4}TaS_2$. a** Structures of $2H$-$TaS_2$, $Cr_{1/4}TaS_2$, and $Cr_{1/3}TaS_2$, with $1 \times 1$, $2 \times 2$, and $\sqrt{3} \times \sqrt{3}$ unit cells. **b** Raman spectra of $Cr_{1/4}TaS_2$, $Cr_{1/3}TaS_2$, and $2H$-$TaS_2$, with Cr superlattice phonon modes highlighted. **c** Selected area electron diffraction of an exfoliated flake of $Cr_{1/4}TaS_2$, with primitive and superlattice reflections indicated. **d** Virtual dark-field images from four-dimensional scanning transmission electron microscopy (4D-STEM) reconstructed using the intensities of $2 \times 2$ and $\sqrt{3} \times \sqrt{3}$ Bragg disks. **e** 4D-STEM diffraction patterns from the regions indicated in (**d**). **f** Field-cooled (FC) and zero-field-cooled (ZFC) DC magnetic susceptibility ($\chi_{DC}$) vs. $T$, measured with a 100 Oe field parallel and perpendicular to $c$. **g** Real part of the AC susceptibility ($\chi'_{AC}$) vs. $T$, measured with an in-plane AC field of 10 Oe. **h** Isothermal magnetization ($M$) vs. $\mu_0 H \perp c$. **i** Arrott plot with linear fit to the 96 K trace.

intercalant disorder as a source of magnetic scattering. In confocal Raman microscopy (with a laser spot size of ~1 $\mu m^2$), we observe a sharp, intense phonon mode at 157 cm$^{-1}$, which we attribute to the $2 \times 2$ Cr superlattice (Fig. 3a, b)[49,50]. A small but sharp feature at 148 cm$^{-1}$ is also present, which matches the $\sqrt{3} \times \sqrt{3}$ superlattice mode observed in the compound $Cr_{1/3}TaS_2$[51]. This suggests that a small amount of $\sqrt{3} \times \sqrt{3}$ ordering with sub-micron in-plane domain sizes is present in bulk crystals of $Cr_{1/4}TaS_2$, and that these $\sqrt{3} \times \sqrt{3}$-containing domains coexist with the dominant $2 \times 2$ superlattice.

Electron diffraction studies on $Cr_{1/4}TaS_2$ unambiguously confirm the existence of ordered $\sqrt{3} \times \sqrt{3}$-containing domains, and furthermore demonstrate nanoscale domains of different superlattices. Selected area electron diffraction (SAED) of a mechanically exfoliated flake in a circular region with a radius of ~ 700 nm shows $2 \times 2$ superlattice reflections, as well as significantly weaker but sharp $\sqrt{3} \times \sqrt{3}$ reflections (Fig. 3c). To investigate the spatial distribution of superlattice domains, we used four-dimensional scanning transmission electron microscopy (4D-STEM). In 4D-STEM, a converged electron beam ( ~ 6 nm under our experimental conditions) is scanned across a two-dimensional (2D) area of the sample, and 2D diffraction data is collected at each probe position[52]. Virtual dark-field images reconstructed using the intensities of $2 \times 2$ and $\sqrt{3} \times \sqrt{3}$ Bragg disks reveal in-plane spatial separation of superlattices (Fig. 3d). The $2 \times 2$ superlattice is dominant over the entire area except for two small regions that resemble pinched triangles. These $2 \times 2$-deficient regions precisely correspond to the brightest regions in the $\sqrt{3} \times \sqrt{3}$ map.

Individual electron diffraction patterns shown in Fig. 3e illustrate the evolution of superlattice order over a 100 nm length scale, showing mixed superlattices with maximal $\sqrt{3} \times \sqrt{3}$ order, weaker $\sqrt{3} \times \sqrt{3}$ order, and exclusive $2 \times 2$ order in moving from a triangular $\sqrt{3} \times \sqrt{3}$-containing region to a $2 \times 2$ region.

**Magnetometry**

The presence of $\sqrt{3} \times \sqrt{3}$ order offers a clue for interpreting bulk magnetometry data on $Cr_{1/4}TaS_2$, which are inconsistent in several respects with the fully compensated AFM structure associated with the $2 \times 2$ Cr superlattice. A sharp rise in the field-cooled DC magnetic susceptibility ($\chi_{DC}$) below 100 K, along with marked bifurcation between zero-field-cooled and field-cooled traces, is suggestive of a minority ferromagnetic (FM) transition (Fig. 3f). The larger $\chi_{DC}$ with the magnetic field applied in-plane is consistent with the expected easy-plane behavior (Fig. S4). Fitting $\chi_{DC}^{-1}$ to the Curie–Weiss law, $\chi^{-1} = (T - \theta_{CW})/C$, yields $C = 1.77(4)$ emu K (mol Cr)$^{-1}$ and $\theta_{CW} = 32(1)$ K (Fig. S5). We then obtain $\mu_{eff} = \sqrt{8C} = 3.76(8)$ $\mu_B$/Cr, close to the theoretical spin-only value of 3.87 $\mu_B$/Cr for Cr$^{3+}$ ($S = 3/2$). The value of $\theta_{CW}$ is positive but considerably smaller than the temperature of the upturn in $\chi_{DC}$, suggesting the coexistence of AFM and FM coupling[53].

The real part of the AC magnetic susceptibility ($\chi'_{AC}$) shows a cusp below 100 K, as well as a broader and more prominent feature with a maximum at about 40 K with a modest frequency dependence (Fig. 3g and Fig. S6). The peak at 40 K is especially pronounced in the imaginary part of the AC susceptibility (Fig. S7). These observations

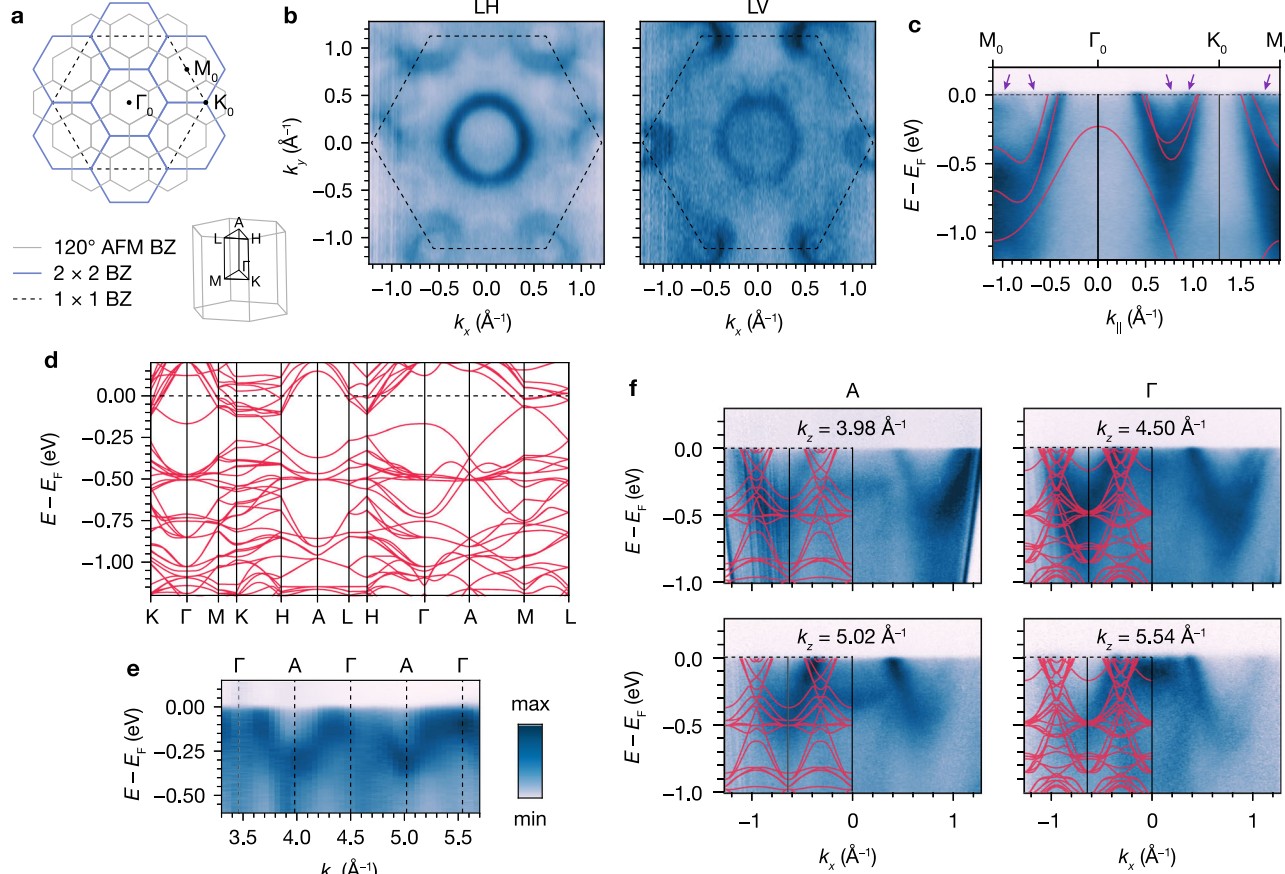

**Fig. 4 | Electronic structure of $Cr_{1/4}TaS_2$. a** Two-dimensional Brillouin zone (BZ) for 120° AFM, 2 × 2 superlattice, and 1 × 1 host lattice, and three-dimensional BZ for 120° AFM, with high-symmetry points labeled. **b** Experimental Fermi surfaces from angle-resolved photoemission spectroscopy (ARPES) measured with linear horizontal (LH) and linear vertical (LV) polarization. Dotted lines indicate the 1 × 1 host lattice BZ. **c** Energy vs. $k_\parallel$ dispersion along $M_0$-$\Gamma_0$-$K_0$-$M_0$. Overlay: density functional theory (DFT) band structure of 2H-TaS$_2$ shifted down by 0.2 eV. Purple arrows indicate replica bands from folding that cross $E_F$. **d** DFT band structure for $Cr_{1/4}TaS_2$ in the 120° AFM phase. **e** Energy vs. $k_z$ dispersion at $k_x = 0$ from the photon energy dependence. **f** Energy vs. $k_x$ dispersions from constant $k_z$ cuts centered at A and Γ. Overlays: DFT band structures along the $K_0$-$\Gamma_0$-$K_0$ direction with $k_z = \pi/c$ and $k_z = 0$ for A and Γ, respectively. All ARPES data were measured at 10 K with 80 eV, LH-polarized photons unless otherwise indicated.

suggest magnetic irreversibility at lower temperatures. We tentatively propose a reentrant spin glass (SG) state below 40 K that is driven by spin disorder arising from competition between AFM and FM exchange interactions[22,23]. This interpretation is corroborated by the slow relaxation dynamics observed in thermoremanent magnetization measurements (Fig. S8 and Table S6)[54].

Partial FM ordering in a minority of the sample is also supported by the isothermal magnetization (M) vs. field traces (Fig. 3h). At 4 K, M with H⊥c shows a small coercive field of 400 Oe (Fig. 3h inset). A detectable hysteresis persists up to 94 K (Fig. S9), above which a slight S-shape remains below 150 K. The magnetization is non-saturating up to fields of 12 T, reaching a maximum value of 0.55 $\mu_B$/Cr at 4 K, which is reasonable for compensated AFM ordering in the majority of the sample. An Arrott plot indicates a first-order FM transition with $T_C = 96$ K (Fig. 3i)[55], in agreement with the AC and DC susceptibility results. The coercive fields observed in the isothermal magnetization with H∥c data are larger, as would be expected for the field applied along the hard axis (Fig. S10), but comparable in magnitude to the AHE hysteresis between 50 and 100 K.

It is known that the fully occupied $\sqrt{3} \times \sqrt{3}$ Cr superlattice in $Cr_{1/3}TaS_2$ exhibits local FM coupling both in- and out-of-plane, with chiral helimagnetic order developing below 140 K because of competition between FM Heisenberg exchange and the Dzyaloshinskii–Moriya (DM) interaction[40,41]. The introduction of

vacancies on the $\sqrt{3} \times \sqrt{3}$ Cr superlattice in the closely related material $Cr_{1/3}NbS_2$ lowers $T_C$ markedly and suppresses the DM interaction, resulting in bulk FM behavior[26,28]. Hence, we attribute the partial FM ordering observed in $Cr_{1/4}TaS_2$ to the minority $\sqrt{3} \times \sqrt{3}$-containing structural domains. The lower $T_C$ of 96 K observed in our samples compared to the reported $T_C$ of $Cr_{1/3}TaS_2$ is consistent with a defective $\sqrt{3} \times \sqrt{3}$ Cr superlattice in the minority FM domains of $Cr_{1/4}TaS_2$, which is expected from the average stoichiometry of our samples.

## Electronic structure

To investigate the effects of Cr intercalation and superlattice ordering on the electronic structure of $Cr_{1/4}TaS_2$, we first consider the symmetry of the 120° AFM and the dominant 2 × 2 structural superlattice relative to the host lattice of 2H-TaS$_2$. The 2D Brillouin zone (BZ) for the 120° AFM BZ is scaled down by $1/\sqrt{3}$ and rotated by 30° relative to the 2 × 2 structural BZ, which is itself scaled down by 1/2 relative to the 1 × 1 host lattice BZ (Fig. 4a). As a result, extensive zone folding is expected in-plane. For the three-dimensional (3D) BZ, the out-of-plane reciprocal lattice vector remains unchanged.

Using angle-resolved photoemission spectroscopy (ARPES), we find that the experimental Fermi surfaces show clear evidence of 2 × 2 reconstruction (Fig. 4b). With linear horizontal (LH) polarized photons (80 eV), we see replica hole pockets at $M_0$ that are folded from $\Gamma_0$. In linear vertical (LV) polarization, we also observe duplication of the

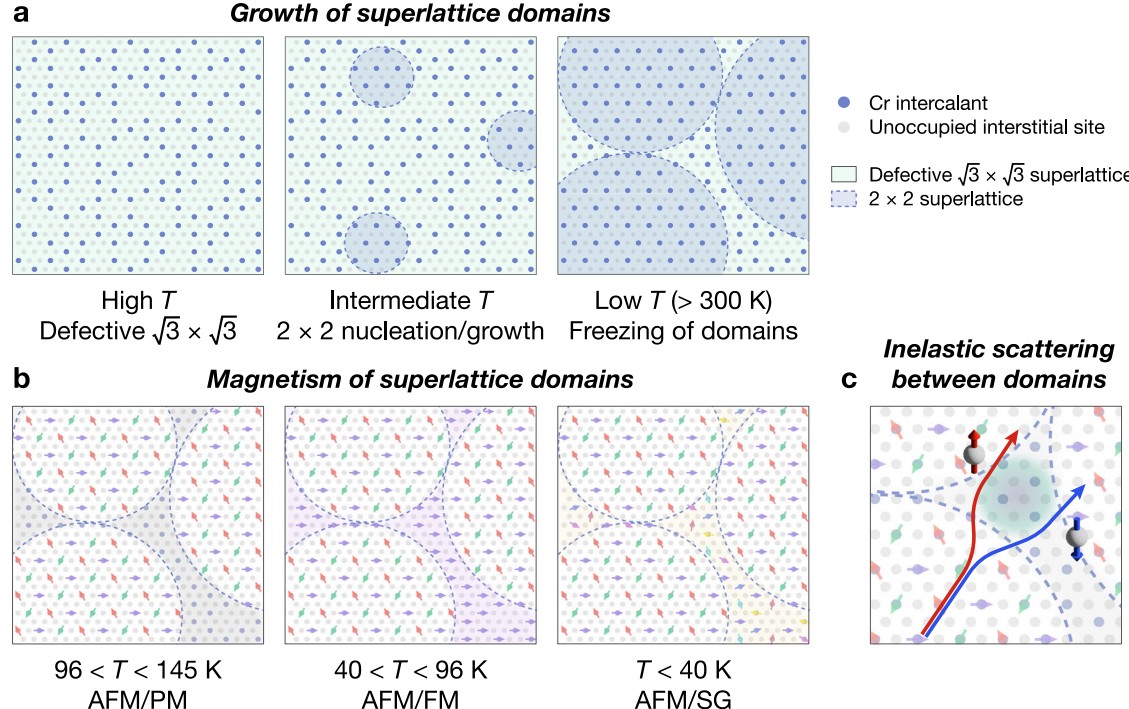

**a** *Growth of superlattice domains*

High *T*
Defective $\sqrt{3} \times \sqrt{3}$

Intermediate *T*
2 × 2 nucleation/growth

Low *T* (> 300 K)
Freezing of domains

- Cr intercalant
- Unoccupied interstitial site
- Defective $\sqrt{3} \times \sqrt{3}$ superlattice
- 2 × 2 superlattice

**b** *Magnetism of superlattice domains*

96 < *T* < 145 K
AFM/PM

40 < *T* < 96 K
AFM/FM

*T* < 40 K
AFM/SG

**c** *Inelastic scattering between domains*

**Fig. 5 | In-plane superlattice domain formation and magnetic phases in Cr$_{1/4}$TaS$_2$. a** Schematic illustration of growth and ordering of Cr intercalant domains at elevated temperatures. Gray dots indicate possible interstitial sites, blue dots indicate Cr, light green shaded regions indicate local $\sqrt{3} \times \sqrt{3}$ order, and light blue shaded regions indicate 2 × 2 order. **b** Proposed magnetic phases in 2 × 2 and $\sqrt{3} \times \sqrt{3}$ structural domains at 145 K and below, comprising antiferromagnetic (AFM), ferromagnetic (FM), paramagnetic (PM), and spin glass (SG) phases. **c** Schematic illustration of inelastic scattering of spin-up and spin-down carriers between AFM 2 × 2 domains and minority domains or domain walls.

hole pockets from K$_0$ at the corners of the 2 × 2 BZ. By fitting the main hole pocket around Γ$_0$ from the LH data, we obtain $k_F = 0.38$ Å$^{-1}$ along Γ$_0$–K$_0$ (Fig. S11), which is intermediate between 2*H*-TaS$_2$ and Cr$_{1/3}$TaS$_2$[51,56].

The ARPES energy vs. momentum dispersion shows further evidence for superlattice-induced electronic reconstruction. As expected, the most prominent bands are consistent with the overlaid density functional theory (DFT) band structure for 2*H*-TaS$_2$ with the Fermi level ($E_F$) shifted down by 0.2 eV (Fig. 4c)[57]. However, clear replica bands are present throughout the BZ that do not correspond to any bands from the unfolded host lattice (indicated by purple arrows in Fig. 4c and Fig. S12). The DFT band structure calculated for the 120° AFM structure of Cr$_{1/4}$TaS$_2$ indicates that the magnetic and structural folding results in several bands crossing $E_F$ (Fig. 4d), thus confirming the experimentally observed features.

We observe a marked $k_z$ dispersion of bands near $E_F$, as shown in Fig. 4e, consistent with the Γ–A dispersion from DFT (Fig. 4d). Constant $k_z$ cuts from the high-symmetry points qualitatively match the DFT band structures for $k_z = 0$ and $k_z = \pi/c$, respectively (Fig. 4f and Fig. S13). This constitutes additional evidence that the Cr superlattice and 120° AFM structure strongly modulate the electronic structure beyond the rigid band picture in a manner consistent with other intercalated TMDs[51,58–61]. More broadly, the ARPES experiments further confirm that the 2 × 2 superlattice is dominant, as we do not observe evidence of electronic reconstruction associated with minority $\sqrt{3} \times \sqrt{3}$-containing structural domains.

## Discussion

From the magnetometry data and geometry of domains observed in 4D-STEM, we propose the following simplified scheme for Cr ordering during crystal growth and the subsequent cooling process (Fig. 5a). At high temperatures, a defective $\sqrt{3} \times \sqrt{3}$ superlattice, which has higher configurational entropy, is more stable. At intermediate temperatures, enthalpically favored 2 × 2 domains nucleate and grow. At lower temperatures, Cr intercalants become immobile due to insufficient thermal energy, thus freezing the superlattice domain configurations at room temperature and below. This implies that if 2 × 2 domains are not given enough time to grow during cooling, defective $\sqrt{3} \times \sqrt{3}$-containing domains will persist between them. The spatial separation of superlattices revealed by 4D-STEM suggests that despite cooling at the relatively slow rate of 20 °C/h and being within experimental error of the perfect Cr$_{1/4}$TaS$_2$ stoichiometry, growth of pure 2 × 2 domains appears to be kinetically limited in our best samples. This picture explains how $\sqrt{3} \times \sqrt{3}$ order can be present in samples with compositions that should favor a 2 × 2 superlattice from purely thermodynamic considerations[32,62], consistent with computational predictions[63].

Accordingly, we propose the following coexistence of magnetic phases in Cr$_{1/4}$TaS$_2$ based on the transition temperatures for the AFM, FM, and SG phases established previously (Fig. 5b). Below 145 K, the majority 2 × 2 domains exhibit 120° AFM order, while the minority $\sqrt{3} \times \sqrt{3}$-containing domains remain paramagnetic (PM). Below 96 K and above 40 K, the $\sqrt{3} \times \sqrt{3}$-containing domains exhibit FM order, and below 40 K, they enter an SG state. We note that our proposed scheme for superlattice domain formation considers only in-plane ordering. From the 4D-STEM data, the $\sqrt{3} \times \sqrt{3}$-containing domains also show 2 × 2 reflections (Fig. 3e), implying that some mixing of superlattices occurs in the out-of-plane direction. It is possible that the structural proximity of superlattices with FM vs. AFM coupling both in- and out-of-plane is responsible for the observed SG behavior.

The magnetic transition temperatures correlate well with changes observed in the AHE. A nonzero AHE emerges below 145 K, coinciding with the onset of AFM ordering. We attribute the negative sign between 100 and 145 K to an extrinsic AHE from inelastic scattering,

specifically from spin defects at domain walls between AFM domains and PM minority superlattice domains (Fig. 5c)[16,64]. The AHE is positive between 40 and 100 K, which corresponds with FM ordering in the minority domains. The complex scaling between the anomalous Hall conductivity ($\sigma_{AHE}$) and the longitudinal conductivity ($\sigma_{xx}$) suggests that different mechanisms are responsible in different temperature regimes (Fig. S14). No clear low-field signature of AHE is observed below 40 K, reminiscent of the disappearance of some components of the AHE in the cluster glass phase of the kagome 120° AFM $Mn_3Sn$[42] (though we note that the kink in the apparent $n_h$ observed below 50 K could suggest higher-field AHE contributions).

These temperature regimes are further supported by the MR behavior, which is generally consistent with disparate magnetic phases coexisting on the nanoscale. The crossover between positive and negative MR and low-field cusps at intermediate temperatures are reminiscent of dilute magnetic semiconductors and magnetically doped topological insulators with clustering of the magnetic species[65,66]. We note also that $Cr_{1/3}NbS_2$ likewise exhibits complex field dependence of the MR and a sign change in the AHE, which have been attributed to field evolution of noncoplanar spin textures[67,68]. In $Cr_{1/4}TaS_2$, although the minority $\sqrt{3} \times \sqrt{3}$-containing domains are defective, local DM interactions or domain wall effects may still lead to spin canting or chiral textures that influence the transport behavior. In general, the qualitative similarity of sign changes in the AHE of several intercalated TMDs suggests that superlattice boundaries may be a general source of skew scattering in these systems, and that further study of superlattice boundary effects on their magneto-transport properties is merited[16,67]. Finally, at low temperatures, the positive, quadratic-like MR and the disappearance of the low-field AHE could indicate that the $\sqrt{3} \times \sqrt{3}$-containing domains become significantly more resistive than the $2 \times 2$ majority domains, resulting in the measured transport behavior corresponding to only the AFM $2 \times 2$ majority domains.

To test our proposed mechanisms, we carried out an analogous set of magnetometry and transport experiments on Cr-deficient crystals with the composition $Cr_{0.23}TaS_2$. The Cr-deficient material clearly shows an AFM transition at 145 K, with no second transition at 96 K. It exhibits a negative $\theta_{CW}$ and a small, linear $M$ vs. $\mu_0 H$, indicating the absence of FM/SG domains (Fig. S15). The Raman spectrum does not show a well-defined $\sqrt{3} \times \sqrt{3}$ peak, and the $2 \times 2$ superlattice mode is slightly broadened and red-shifted, consistent with a defective super-lattice (Fig. S16). Critically, we observe zero detectable AHE at 200 K and below (Fig. S17). The Cr-deficient material has a *RRR* value of 1.6, almost an order of magnitude lower than $Cr_{1/4}TaS_2$, which we attribute to increased scattering off of defects in the Cr superlattice. The higher carrier concentration is consistent with larger hole pockets due to the lower Cr stoichiometry. We also observe lower mobility and lower MR, which are additionally consistent with lower crystal quality and the absence of FM/SG domains. These findings indicate that the combined presence of a well-ordered and fully occupied $2 \times 2$ Cr superlattice and minority $\sqrt{3} \times \sqrt{3}$-containing domains is required for the rich magneto-transport behavior observed in $Cr_{1/4}TaS_2$.

In summary, $Cr_{1/4}TaS_2$ is a metallic antiferromagnet with a non-collinear 120° AFM ground state. High-quality crystals with a well-ordered $2 \times 2$ Cr superlattice nevertheless also contain minority domains with local $\sqrt{3} \times \sqrt{3}$ order. These samples host complex magnetotransport phenomena below the AFM $T_N$, including temperature-dependent sign changes in AHE and MR. Interactions between the majority AFM phase and minority PM/FM/SG domains engender the observed transport responses through scattering between the disparate superlattices. Altogether, this work illustrates that growth conditions (in addition to composition) must be taken into account to determine superlattice identity and microstructure. Our results corroborate a growing body of literature indicating that subtle differences in crystallographic order in intercalated TMDs can be

harnessed towards realizing a wide array of magnetic and electronic properties[24,25,28,31,69–71].

The possibility of engineering disparate magnetic phases by controlling the geometry of superlattice domains, as well as the topology of domain walls, is an intriguing prospect and worthy of further exploration[72,73]. In addition, intercalated TMDs might be promising platforms to study analogues of the three- and four-state Potts universality classes (corresponding to 2D $2 \times 2$ and $\sqrt{3} \times \sqrt{3}$ ordering, respectively) in three dimensions[74]. More generally, our work points to the importance of using sensitive and local probes to investigate the possibility of disorder or inhomogeneity in intercalation compounds with more than one stable superlattice. Further tailoring of scattering interactions via optimization of superlattice nucleation and growth may lead to the development of new materials exhibiting electrical transport responses relevant to spintronic devices.

## Methods
Single crystals of $Cr_{1/4}TaS_2$ were grown from Cr (powder, 99.97%, Alfa Aesar), Ta (powder, 99.98%, Alfa Aesar), and S (powder, 99.999%, Acros Organics) using $I_2$ (99.999%, Spectrum Chemicals) as a transport agent. The constituent elements in a 0.33:1:2.05 molar ratio were sealed in an evacuated fused quartz ampoule (14 mm inner diameter, 1 mm wall thickness, 25 cm long) along with $2 \, mg/cm^3$ of $I_2$. The ampoule was placed in a two-zone furnace with the hot zone maintained at 1100 °C and the cold (growth) zone maintained at 1000 °C for 200 h, after which both zones were cooled down to room temperature over the course of 50 h (about 20 °C/h). Hexagonal plate-shaped crystals were obtained, with lateral dimensions of several mm and thicknesses of 1–2 mm. Single crystals of $Cr_{1/3}TaS_2$ were grown in an analogous fashion using a 0.47:1:2.1 molar ratio. Single crystals of $Cr_{0.23}TaS_2$ were grown in an analogous fashion using a 0.30:1:2.02 molar ratio, $2.3 \, mg/cm^3$ of $I_2$, and a 1100 °C hot zone and 950 °C cold zone. Single crystals of $2H$-$TaS_2$ were obtained from HQ Graphene and mechanically exfoliated onto $SiO_2$/Si.

Single-crystal X-ray diffraction data were collected on a Rigaku XtaLAB P200 with Mo K$\alpha$ radiation at 295 K. Data reduction and scaling and empirical absorption correction were performed in CrysAlis Pro. Structures were solved by direct methods using SHELXT[75] and refined against $F^2$ on all data by full-matrix least squares with SHELXL[76] using the ShelXle graphical user interface[77]. Reconstructed scattering planes were generated from the frames in CrysAlis Pro. Energy dispersive X-ray spectroscopy data were acquired on a FEI Quanta 3D FEG scanning electron microscope with an accelerating voltage of 20 kV.

Single-crystal neutron diffraction measurements were conducted on $WAND^2$ at the High Flux Isotope Reactor (Oak Ridge National Laboratory) with an incident wavelength of 1.486 Å. A single crystal was mounted on an aluminum rod with GE Varnish in the *HHL* scattering geometry. The data were integrated using Mantid Workbench[78]. Representational analysis was carried out using SARAh[45], and structural refinement was performed using FullProf[79].

Heat capacity, electrical transport, and magnetometry measurements were carried out in a Quantum Design Physical Property Measurement System Dynacool equipped with a 12 T magnet. For heat capacity measurements, single crystals were affixed to the stage using Apiezon N grease. For electrical transport measurements, cleaved single crystals with thicknesses of 30 $\mu$m or less were affixed using GE Varnish and contacted using silver paint and gold wire. Typical dimensions were on the order of $50 \times 100 \times 20 \, \mu$m. Measurements were conducted using Stanford Research Systems SR830 lock-in amplifiers by applying a 5 mA AC current (17.777 Hz) and measuring the transverse and longitudinal voltages in typical four-probe or Hall configurations. $\rho_{xx}$ and $\rho_{yx}$ data were symmetrized and anti-symmetrized, respectively, according to the following equations (with → indicating the forward sweep direction, and ← indicating the

negative sweep direction):

$$\rho_{yx,\,symm}^{\rightarrow}(\mu_0 H) = \frac{1}{2}[\rho_{yx}^{\rightarrow}(\mu_0 H) - \rho_{yx}^{\leftarrow}(-\mu_0 H)] \tag{1}$$

$$\rho_{yx,\,symm}^{\leftarrow}(\mu_0 H) = \frac{1}{2}[\rho_{yx}^{\leftarrow}(\mu_0 H) - \rho_{yx}^{\rightarrow}(-\mu_0 H)] \tag{2}$$

$$\rho_{xx,\,symm}^{\rightarrow}(\mu_0 H) = \frac{1}{2}\left[\rho_{xx}^{\rightarrow}(\mu_0 H) + \rho_{xx}^{\leftarrow}(-\mu_0 H)\right] \tag{3}$$

$$\rho_{xx,\,symm}^{\leftarrow}(\mu_0 H) = \frac{1}{2}\left[\rho_{xx}^{\leftarrow}(\mu_0 H) + \rho_{xx}^{\rightarrow}(-\mu_0 H)\right] \tag{4}$$

Magnetoresistance (MR) data were calculated according to the following equation:

$$MR(\mu_0 H)(\%) = \left[\frac{\rho_{xx}(\mu_0 H)}{\rho_{xx}(0)} - 1\right] \times 100\% \tag{5}$$

$\rho_{AHE}$ and MR data were denoised with a Savitzky–Golay filter; a notch filter was applied to MR data to remove low-frequency noise on the order of 0.001 Hz. Magnetometry measurements were performed using the Vibrating Sample Magnetometer option for DC measurements, and the AC Measurement System II option for AC measurements, with crystals affixed to quartz paddles or brass holders using GE Varnish.

Confocal Raman microscopy data were collected on a Horiba LabRAM HR Evolution with an ultra-low frequency filter using 633 nm laser excitation and powers between 1 and 8 mW.

Plan-view transmission electron microscopy (TEM) imaging of a ~ 50 nm-thick $Cr_{1/4}TaS_2$ flake was performed using an FEI TitanX operating at 80 keV. Selected area electron diffraction (SAED) and four-dimensional scanning transmission electron microscopy (4D-STEM) were acquired in the same sample region. Samples were prepared via a dry transfer method utilizing a poly(bisphenol A carbonate)/polydimethylsiloxane polymer stamp[17]. In this process, $Cr_{1/4}TaS_2$ flakes were mechanically exfoliated using Kapton tape onto $SiO_2$/Si, and were then transferred onto a 200 nm silicon nitride holey TEM grid (Norcada) that had been treated with $O_2$ plasma for 5 min immediately prior to stacking. For the SAED experiments, a 40 μm diameter aperture was used, defining a selected diameter of ~ 720 nm. On the other hand, 4D-STEM was acquired with a 0.55 indicated convergence semi-angle yielding a ~ 6 nm converged electron probe size. The acquired 4D-STEM data were analyzed using the py4DSTEM Python package[80]. Peak detection algorithms identified the $H$-$TaS_2$ Bragg peaks, which were used to calculate the reciprocal lattice vectors of the $H$–$TaS_2$ structure. Based on their symmetry relations to these vectors, the $2 \times 2$ and $\sqrt{3} \times \sqrt{3}$ Cr superlattice reciprocal vectors were determined. Virtual apertures were then constructed separately for each superlattice by masking all regions of the diffraction patterns except for the corresponding Cr superlattice peak areas. These apertures were applied to the 4D-STEM data to extract the integrated intensities of the two distinct superlattices at each probe position.

Angle-resolved photoemission (ARPES) measurements were carried out on beamline 4.0.3 (MERLIN) of the Advanced Light Source (Lawrence Berkeley National Laboratory) equipped with a Scienta Omicron R8000 hemispherical electron analyzer. Crystals were cleaved in situ under high vacuum (base pressures of $5 \times 10^{-11}$ Torr or less) by carefully knocking off alumina posts affixed to the top surface using silver epoxy. Photon energy-dependent measurements were conducted between 30 and 124 eV, and momentum conversion was carried out using an inner potential ($V_0$) of 8 eV. The primary datasets were collected at $h\nu = 80$ eV, close to $\Gamma_0$ as determined by the photon

energy dependence. Data analysis was carried out using the PyARPES software package[81].

First-principles calculations based on density functional theory (DFT) were performed using the Vienna Ab Initio Simulation Package (VASP)[82,83] and the open source plane-wave code QuantumEspresso (QE)[84]. The projector-augmented wave (PAW) methods[85] implemented in VASP was used with a kinetic energy cutoff of 400 eV. The optimized norm-conserving Vanderbilt (ONCV) pseudopotentials from the PseudoDojo project[86,87] were applied for calculations using QE with a kinetic energy cutoff of 86 Ry. The lattice constants ($a = 6.584(2)$ Å, $c = 12.015(2)$ Å) of $Cr_{1/4}TaS_2$ were taken from a single-crystal X-ray diffraction measurement at 100 K. A $\Gamma$-center $2 \times 2 \times 2$ k-mesh was used to sample the Brillouin zone for $Cr_{1/4}TaS_2$, and $8 \times 8 \times 2$ k-mesh was used for $TaS_2$. The exchange-correlation interaction was described by the Perdew-Burke-Ernzerhof (PBE) functional[88]. An effective on-site Coulomb interaction of 4 eV on Cr atom was employed within DFT+$U$ of the Dudarev scheme[89]. The magnetic ordering in $Cr_{1/4}TaS_2$ was taken as the 120° noncollinear antiferromagnetic structure using a magnetic unit cell shown in Fig. 1e. VASP was used for the calculations of band structure and magnetic properties in $Cr_{1/4}TaS_2$, and QE was used for the calculation of band structure in $TaS_2$.

## Data availability
Crystallographic data generated in this study have been deposited at the Cambridge Structural Database under the deposition number 2388468. Raw data and processed data generated in this study are available on Zenodo at https://doi.org/10.5281/zenodo.15522287[90].

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

## Acknowledgements

We thank Sae Hee Ryu for assistance with ARPES measurements. This material is based upon work supported by the U.S. National Science Foundation, under award no. 2426144 (D.K.B.). A portion of this research used resources at the High Flux Isotope Reactor, a DOE Office of Science User Facility operated by the Oak Ridge National Laboratory. The beam time was allocated to WAND$^2$ on proposal number IPTS-30492.1. This research used resources of the Advanced Light Source, which is a DOE Office of Science User Facility under contract no. DE-AC02-05CH11231. Work at the Molecular Foundry, LBNL, was supported by the Office of Science, Office of Basic Energy Sciences, the U.S. Department of Energy under Contract no. DE-AC02-05CH11231. The computational research was primarily supported by the NSF through the University of Wisconsin Materials Research Science and Engineering Center (DMR-2309000). L.S.X. acknowledges support from the Arnold and Mabel Beckman Foundation through an Arnold O. Beckman Postdoctoral Fellowship (award no. 51532). B.H.G. was supported by the University of California Presidential Postdoctoral Fellowship Program (UC PPFP), Schmidt Science Fellows in partnership with the Rhodes Trust, and the Max Planck Society. O.G. acknowledges support from a NSF Graduate Research Fellowship Grant DGE 1752814. S.H. acknowledges support from the Blavatnik Innovation Fellowship.

## Author contributions

L.S.X. and D.K.B. conceived of the study. L.S.X. synthesized samples and carried out transport and energy dispersive X-ray spectroscopy experiments. L.S.X., O.G., and S.S.F. collected and analyzed SCXRD data. M.D.F., S.S.F., and L.S.X. collected and analyzed neutron diffraction data. W.F. and K.L. carried out symmetry analysis and first-principles DFT calculations. L.S.X., M.P.E., C.M., and S.H. collected and analyzed Raman data. S.H. collected and analyzed electron diffraction data; B.H.G. and I.M.C. provided analysis and interpretation. C.M., L.S.X., and S.S.F. collected and analyzed magnetometry and heat capacity data. L.S.X., J.P.D., S.S.F., and M.P.E. collected and analyzed ARPES data. D.K.B. and Y.P. supervised the project. L.S.X. wrote the manuscript with input from all authors.

## Competing interests

The authors declare no competing interests.
