## [Peer Review File · Nature Communications]

Anomalous Hall effect from inter-superlattice scattering in a noncollinear antiferromagnet

Corresponding Author: Professor Daniel Bediako

Version 0:

Reviewer comments:

Reviewer #1

(Remarks to the Author)

This manuscript addresses the kinetic control of superlattices in terms of domain structure and its consequences on the structural, magnetic and electrical properties. In particular, CrTaS alloy is considered and characterized by a large variety of experimental techniques. The main message of the manuscript is how the variation of local ordering can be used to tune magnetotransport in an antiferromagnet, with implications for designing transport responses via deliberate superlattice engineering without changes in chemical composition.

This work is highly significant in the field of research on superlattices for antiferromagnetic-based spin-electronics, and contains definitely original results to be worth for a publication. The discussion and the conclusion are strongly supported by a large amount of experimental data obtained by different techniques, with a methodology very appropriate. English is correct, and the figures are clear.

The only points where the manuscript should be improved, to my personal opinion, regard the data interpretation by theoretical models.

DFT results are presented in the text, and partially in the methods and in supplementary, but a comprehensive discussion is lacking. How strongly can they be used to confirm the experimental data? How robust is the interpretation? Please comment on this. I would also suggest to increase the dimensions of Figs.4d and f, that are too small to look at the calculated band structure and its superposition on experimental data.

At page 7, the authors introduce the Berry curvature to explain the AHC. It is not clear if they do it by calculation, or just take literature results ("can be evaluated"). Please comment on this, and add more information on the AHC interpretation by Berry curvature: why and how it is done, and which are the results.

A more general comment: both data and experimental techniques are so many, in the paper, that at a certain point it results difficult to follow the logic behind the work. I would suggest to split the Results section in some sub-sections with titles addressing the type and/or technique of characterization.

Reviewer #2

(Remarks to the Author)

The paper by Lilia S. Xie et al. addressed timely topic of different phases and their impact on the properties of the system in the TM-M-dichalcogenide group. These materials have been studied in the 70's and 80's (e.g. B. van Laar et al., S.T.T. Parkin et al.) and now experience a renaissance with the discovery of van der Waals materials and their engineering and topological aspects of the band structure. One of such interesting materials is Cr_{1/3}TaS₂, in which formation of chiral/heli magnetic ordering with phase transition at about 140K-150K was observed (doi.org/10.1063/5.0166333, [doi:10.1088/1742-6596/746/1/012061](https://doi.org/10.1088/1742-6596/746/1/012061), doi.org/10.1002/aelm.202100424). The Authors of this submission create yet a new phase: Cr_{-x}TaS₂ composed of these elements, with x=1/4 and a new Cr sub-lattice (2 x 2) clearly resolved in the electron diffraction studies, incorporated in the known Cr_{1/3}TaS₂ phase with a pattern ($\sqrt{3} \times \sqrt{3}$). They evidence an intriguing anomalous Hall effect, that changes sign with temperature, which is a main discovery of the paper. Numerous methods are used to fully characterize the system, from calorimetric studies, neutron, X ray and electron diffraction, Raman scattering, magnetization and ARPES.

While I find the results interesting, timely and original, some parts of the paper are confusing and need reformulation and/or more in-depth discussion. Let me put my remarks in points, in a chronological order.

1. Abstract states “Chemical composition is commonly considered to be the primary determinant of superlattice identity, especially in intercalation compounds. Here, we find that, contrary to this conventional wisdom, kinetic control of superlattice growth leads to the coexistence of disparate domains within a compositionally “perfect” single crystal”. I find this message misleading. I do not see the points where Authors put that common wisdom in question. Instead, they rely on it: it is the correct chemical content that is fundamental for this research, it is a new phase, thermodynamically possible, which was demonstrated. Instead of contrast it would be more appropriate to speak about “an added value”.
2. The claim of “microscopic patterning” in the context of self-organized sample with disorder is misleading. That would be authorized if the Authors present a series of samples with different cooling rates to tune proportions between two phases observed and demonstrate control of this new phase formation. Or there is any patterning that I missed here? Perhaps sample post-growth processing?
3. Referring to the electrical transport data, in $\rho_{xx}(T)$ there is a kink (it is not a “sharp drop”) evidencing magnetic phase transition. It is ascribed to transition to the antiferromagnetic state. What are the arguments supporting that it is in fact an antiferromagnetic phase in $\text{Cr}_{-1/4}\text{TaS}_2$ that develops below 145 K? Was there any magnetic anisotropy observed related to this phase? While the sample contains two phases, perhaps this transition occurs in $\text{Cr}_{-1/3}\text{TaS}_2$ phase, especially since literature data evidence phase transition at 140-150 K in this material. Do the Authors have an idea why the second phase transition is not visible in the $\rho_{xx}(T)$ data? Also, why should this high temperature phase be absent below $\sim 100\text{K}$ and not contribute to the AHE? A comment should be provided. As for the ferromagnetic phase, present below 100K, it is ascribed to the $\text{Cr}_{1/3}\text{TaS}_2$ phase with a pattern $(\sqrt{3} \times \sqrt{3})$ and magnetization in the c plane (Fig.3h). This phase is described as responsible for the appearance of the AHE signal. Why would the in-plane magnetization lead to the AHE (when B is applied perpendicular to plane) in spite of the in-plane magnetization? The evolution of M vs B-out-of-plane is shown in Fig.S7 and here the coercive field is much lower than this observed in the AHE, putting some doubts to the interpretation.
4. Authors study “in-plane Hall resistivity ρ_{xy} ”. Typically, it is the ρ_{yx} component of the resistivity tensor that is called Hall resistivity [e.g. C. L. Chien, C. R. Westgate, Hall Effect and its Applications]. Also, what does “in-plane” refers to? Is magnetic field applied in the plane of the layer? Basing on the information provided, it seems that magnetic field is oriented along c axis and it is the usual Hall effect.
5. Referring to the interpretation of the Hall resistivity: when anomalous Hall signal is present, it is often impossible to decompose the ordinary and anomalous components and to extract the values of concentration (the anomalous one happens to be present even at high field). This is the case here: the kink in the $n(T)$ dependence does not reflect change in concentration, but comes from the AHE contribution.
6. Discussing Anomalous Hall Effect the Authors rule out an intrinsic effect, related to the Berry curvature, as an origin of the AHE signal $T=145\text{K}$, indicating that this component is zero. I think that this fact should be justified in a more elaborate way or references should be given to support this claim. The mirror symmetry is present only when the magnetic moments are flipped (so when time reversal symmetry is broken at the same time). It should be commented. Also, it is important to notice that S-O effects also contribute to the Berry curvature. Perhaps presented calculations may be extended to demonstrate vanishing Berry curvature integrals (at $B=0$)?
7. Referring to the AHE conductivity, it is an extra component of the conductivity: $\sigma_{yx} = \sigma_{\text{AHE}} + \sigma_{yx}(\text{ordinary})$. The Hall conductivity transforms as $\sigma_{yx} = \rho_{yx} / (\rho_{xx}^2 + \rho_{yy}^2)$, where ρ_{yx} is experimentally found. My doubts concern the way the AHE conductivity was calculated (replacing simply ρ_{xy} by AHE). In my opinion it is the total sigma that should be calculated and then anomalous component should be identified.
8. I find the names: bulk and minority domains (e.g. in the abstract) and the opposition between them, confusing. It should be clearly stated whether 2×2 and $(\sqrt{3} \times \sqrt{3})$ domains are surface domains of bulk ones. It is not clear in which parts the Authors study few layer exfoliated flakes and where bulk samples.
9. I do not quite understand the role of the control sample, with $x=0.23$ and Cr deficiency. It has lower crystal quality, higher concentration, but still two phases with different Cr arrangements (and some Cr missing) evidenced in the Raman scattering. One would expect the AHE to be present, when it is absent for both. Was sample geometry the same of transport experiments? Why “higher carrier concentration is consistent with the absence of FM domains” – that needs more justification. The possible way I see is though Berry phase dependence of Fermi level position, but Authors rule out this mechanism.
10. In Fig.3 f the curves for $H \parallel c$ are not visible at all – why not use right vertical scale?
11. Instead of “Zero field hysteresis of less than 1T” one should say that coercive field is about 1T.
12. In Fig.S4 the frequency evolution of the peak at higher temperatures should be shown.
13. Comment in the SI in the crystal growth section (see the Supplementary Information for more details on crystal growth and compositional characterization) is missing. Also, the description of the studied samples is not sufficient, are they bulk or exfoliated samples? For transport experiment, how were the hallbars prepared, what were the dimensions? In addition to the Tables (s1 and s2) showing the presence of the a given phase, the XRD spectra should be presented in the supplementary. While in log scale, they evidence the lack of parasitic phases which could contribute to both magnetic and transport data

Overall, I think that the data and physics of this material is very interesting. However, being interesting it is also truly complex and more precise explanations are often needed for the reader to comprehend and appreciate the content. Therefore I would recommend the Authors to address the points I raised.

Reviewer #3

(Remarks to the Author)

In this work, the authors discuss the growth and characterization of Cr intercalated TaS₂ (Cr_{1/4}TaS₂). They identify a magnetic transition at ~145 K via a weak heat capacity signature and the magnetic structure via neutron diffraction as a 120 degree antiferromagnetic (AFM) phase. The very unusual properties of the material manifest in the presence of an anomalous Hall effect (AHE), common in ferromagnetic (FM) systems, but generally less common in antiferromagnetic ones owing to the lack of overall time reversal symmetry breaking. They further characterize slight differences in the microscopic structures and ordering of the Cr intercalants that via Raman and transmission electron microscopy. The rest of the article constructs a toy model scenario around this Cr intercalated microstructure about coexisting AFM/FM domains that could potentially give rise to an AHE and the unusual magneto transport. As the anomalous Hall effect is a fascinating phenomenon that elucidates the relation between the topology of Bloch wavefunctions and their transport/scattering processes, the work in this manuscript could likely be important in this regard. The experimental work in growth and characterization is extensive, and the authors' dedication shows in the figures and text of the manuscript. There is perhaps a lack of detailed quantitative theory/justification of how the processes give rise to the AHE and magneto transport, especially the sign change of the AHE conductivity with temperature and something conclusive of how it scales with longitudinal conductivity, but perhaps this work indeed will spark future studies to address this question.

I recommend the article for publication once the authors address some detailed concerns as listed below.

1. Upon my initial reading of the abstract, I was confused about what "domain" meant in this context, as the title leads me to believe that AFM domain exist and that domain wall between them are responsible for this anomalous Hall effect. I initially thought it was referring to ordered magnetic domains (ie AFM), but was actually in reference to the Cr intercalants forming a 2×2 and $\sqrt{3} \times \sqrt{3}$ superlattice, which of course plays a key role in the different magnetic phases discussed later in the paper. The abstract should be clear about the type of domains studied - consider mentioning that the Cr intercalants form such 2×2 and $\sqrt{3} \times \sqrt{3}$ superlattice and mention how this is determined (via transmission electron microscopy). In addition, the abstract is a bit lean on what experimental techniques are involved (I count heat capacity, magnetotransport, Hall effect, Raman, neutron scattering, and magnetization) of which the authors should take pride in and mention a few.
2. On page 5 there is a sentence that reads "The in-plane Hall resistivity (ρ_{xy}) is dominated by the ordinary Hall effect" – this needs to be reworded. Although it is clear from the context of this paper that voltage, current and field are all perpendicular to each other, the concept of "in-plane" Hall effect is a real phenomenon that can manifest with the field in the xy plane. See for example: J. Zhou et al., Nature 609, 46-51 (2022) or even your own reference 66 Mayoh et al.
3. On the MR of Fig. 2f, there appears to be hysteresis in some of the MR data (ie 120 K data presented in inset), although perhaps not believable for the 50 K data. However, it is not clear which of the traces correspond to the upswing and the downswing. Also, the 120 K MR data in Fig. 2f appears to show hysteretic behavior persisting up to ± 2 T, but the 120 K AHE in Fig. 2d exhibits non hysteretic behavior (expected since FM onsets at 98 K) and saturates at a modest field of perhaps ± 0.5 T. Could the authors comment on this discrepancy and compare/contrast saturation fields between MR and AHE, as well as presence/absence of hysteretic behavior between MR and AHE.
4. Related to the above point, is it clear where the current is flowing as the temperature is varied and the nature of the $\sqrt{3} \times \sqrt{3}$ minority magnetic state changes from PM to FM to spin glass (SG)? Presumably, the disappearance of AHE at low temperatures could be attributed not to the lack of AHE in the SG phase, but perhaps simply because it is much more resistive than the 2×2 AFM majority regions and we no longer obtain boundary scattering as the majority of the current just flows around the SG regions through the network of 2×2 AFM regions. The large positive (quadratic-like?) MR at 10 K relative to the negative MR at higher temperatures seems consistent with this picture as SG typically have negative MR (seen in your higher temperature data). Also, a minor issue: clarify if the antisymmetrization and symmetrization of AHE and MR, was done by mixing upsweeps and downsweeps? ie $\rho_{xy_up}(+H) - 0.5(\rho_{xy_up}(+H) - \rho_{xy_down}(-H))$ and $\rho_{xx_down}(+H) - 0.5(\rho_{xx_down}(+H) + \rho_{xx_up}(-H))$. I believe this is the only way hysteretic behavior can appear in your presented AHE and MR data in Fig. 2, but please mention this explicitly in the methods section towards the end of the paper.
5. The sentence at the end of page 5 "The saturation magnitude of ρ_{AHE} increases with increasing T in this regime" is confusing, because one typically assumes that ρ_{AHE} is the zero-field extrapolated finite component and so the term "saturation" may be misleading. Also, given this definition, the vertical axis of Fig. 2c (ρ_{AHE}) is also misleading. The authors should just clarify the definitions and delineate extracted quantities from fit parameters.
6. Regarding Figure 5 and the growth/high-T picture, although exact identification of the 2×2 and $\sqrt{3} \times \sqrt{3}$ formation temperatures would likely involve differential scanning calorimetry and/or thermogravimetric techniques, I believe faith in the authors' expertise is warranted and they can be forgiven for speculating the temperature ranges at which these metallurgical processes occur. This would allow them to replace/clarify the vague "High T" with something like $\sim 700 \text{ C} < T < 800 \text{ C}$, and similar for Intermediate T and Low T. Changes to the main text are also warranted in order to avoid the rather abrupt transition from previous discussion and crystal growth regime. Also take care in relabeling Fig. 5a temperature regimes so

as to avoid confusing the reader with the previous “low T” data, as well as the pics in panel b), where one may think “high T” in panel a) corresponds with 98-145K in panel b).

7. Finally, regarding the discussion paragraph on page 13 and 14 talking about the AHE, I think this should be expanded on, as it is the key result of this work. The sign change of the AHE is quite interesting and points to something drastic happening with a change in transport through the $\sqrt{3} \times \sqrt{3}$ regions of the material at the PM/FM transition. As the $\sigma_{AHE_{xy}}$ scales linearly with σ_{xx} (Fig. S11) this system is perhaps sensitive to skew-scattering processes, not surprising given the magnetic scattering presumably present at this domain boundary. Its interesting that the magnitude of the apparent slope of $\sigma_{AHE_{xy}}$ with σ_{xx} is about the same before and above the PM to FM transition (ie the sign change), suggesting that the actual microscopic nature of the scattering process does not change nor the band structure/density of states (determined by the 2×2 band structure). Perhaps this is a coincidence, but perhaps a believable postulation can be drawn about a rather unusual change in orientation of the spins responsible for the skew scattering must change direction. Are the authors aware of other superlattice to superlattice boundaries of magnetic phases and local magnetic structures/defects that could possibly be present at this domain boundary? Any other examples of sign changes of the AHE in similar or dissimilar systems should also be cited.

Reviewer #4

(Remarks to the Author)

Version 1:

Reviewer comments:

Reviewer #1

(Remarks to the Author)

For what concerns the points I raised, authors made an excellent work to clarify them (in particular the AHC conductivity and its explanation by Berry curvature), improving the scientific clarity and content of the paper, that to my opinion can be published as is.

Reviewer #2

(Remarks to the Author)

Reviewer #3

(Remarks to the Author)

The authors have done a wonderful job of addressing all concerns. I recommend proceeding to publication.

Reviewer #4

(Remarks to the Author)

Dear Editor, dear Authors,

I like the comments and improvements introduced by Authors, in reply to my comments and the ones of the other referees.

Still, for the sake of clarity I would advise to use different name for the areas of different structure than “domain”. Having read the improved paper, I think that for a paper tackling with different structural and magnetic phases, it would be better to avoid word “domain” having very precise meaning in magnetism, both ferromagnetic and antiferromagnetic, being central for this paper, in order to avoid any ambiguities. Otherwise, I am afraid the reader might be confused that the main observations refer to the interactions between ferromagnetic and antiferromagnetic domains, while they do not. Moreover, there is no proof that the areas of a given structure and magnetic order (called domains) are indeed single domains.

I would also opt for slightly less firm conclusion, while speaking about the very interesting finding of AHE changing sign as a function of temperature, for sample having two phases of distinct magnetic orders. The abstract states: “we demonstrate that $\text{Cr}_{1/4}\text{TaS}_2$ is a bulk noncollinear antiferromagnet in which **scattering between majority and minority superlattice domains engenders complex magnetotransport** below the Néel temperature, including an anomalous Hall effect “. In my opinion there is no experimental proof that this is this scattering that is responsible for the AHE. To my understanding, at this stage this claim is rather a hypothesis that needs future verification (larger number of samples with various content of 2×2 and $\sqrt{3} \times \sqrt{3}$ – phases). The Authors seem to be aware of numerous options possible in this complex system, as discussed in the central paragraph, p.15 of the newest version. In my opinion, the control experiment presented, does not test really test the proposed mechanism, as it bases on interactions while one ingredient of the proposed mechanism is “not conclusively present” (as discussed in the response to ref#2 (my) remark No 9.).

Sincerely,

Referee #2

Dear Dr. Bladwell,

Thank you for your consideration of our manuscript “Anomalous Hall effect from inter-superlattice scattering in a noncollinear antiferromagnet” (manuscript ID NCOMMS-24-70569-T) for publication in *Nature Communications*. We are grateful to the reviewers for their thoughtful and constructive comments, and address them point-by-point below.

Reviewer #1 (Remarks to the Author):

This manuscript addresses the kinetic control of superlattices in terms of domain structure and its consequences on the structural, magnetic and electrical properties. In particular, CrTaS alloy is considered and characterized by a large variety of experimental techniques. The main message of the manuscript is how the variation of local ordering can be used to tune magnetotransport in an antiferromagnet, with implications for designing transport responses via deliberate superlattice engineering without changes in chemical composition.

This work is highly significant in the field of research on superlattices for antiferromagnetic-based spin-electronics, and contains definitely original results to be worth for a publication. The discussion and the conclusion are strongly supported by a large amount of experimental data obtained by different techniques, with a methodology very appropriate. English is correct, and the figures are clear.

The only points where the manuscript should be improved, to my personal opinion, regard the data interpretation by theoretical models.

DFT results are presented in the text, and partially in the methods and in supplementary, but a comprehensive discussion is lacking. How strongly can they be used to confirm the experimental data? How robust is the interpretation? Please comment on this.

Response: We are very grateful to the reviewer for their positive evaluation and feedback. Regarding the DFT results, we have compared the calculated band structure with the AFM magnetic structure to the experimental ARPES data. We find that the ARPES results are largely consistent with the DFT results. Importantly, we experimentally observe both the band folding expected from the 2×2 superlattice, as well as the magnetic unit cell, which lends credence to our assertion that the 2×2 superlattice is the dominant mode of intercalant ordering in our samples. However, because of the large number of bands crossing the Fermi level (as seen in the experimental data and the DFT band structure), it is difficult to make more definitive assignments of bands without a more detailed ARPES study (including more polarization dependence and photon energy dependence scans, along with orbital-projected band structure calculations).

As addressed in more detail in the response to the next point that this reviewer raises, we have added DFT calculations of the anomalous Hall conductivity, which we believe further confirms our experimental transport and magnetometry data.

Revision: We have edited a sentence in the Results section to state explicitly that the DFT band structure calculations confirm the major features observed in the experimental ARPES experiments.

The DFT band structure calculated for the 120° AFM structure of $\text{Cr}_{1/4}\text{TaS}_2$ indicates that the magnetic and structural folding results in several bands crossing E_F (Figure 4d), thus confirming the experimentally observed features.

I would also suggest to increase the dimensions of Figs.4d and f, that are too small to look at the calculated band structure and its superposition on experimental data.

Response: We thank the reviewer for the advice to improve the legibility of Figure 4. We agree that increasing the sizes of these panels will make the band structure clearer and also clarify the comparison with experimental data.

Revision: We have rearranged the panels in Figure 4 and increased the sizes of panels d and f. The updated figure is included below:

At page 7, the authors introduce the Berry curvature to explain the AHC. It is not clear if they do it by calculation, or just take literature results (“can be evaluated”). Please comment on this, and add more information on the AHC interpretation by Berry curvature: why and how it is done, and which are the results.

Response: We thank the reviewer for their feedback. For the theoretical part, we have added a section in the Supplementary Information (SI) to show that the intrinsic anomalous Hall conductivity (AHC) is zero, based on both symmetry analysis and DFT calculation. Intrinsic AHC is determined by the Berry curvature integrated over the Brillouin zone (BZ). If there is a symmetry operation R acting on the Berry curvature so that $R\Omega(\mathbf{k}) = -\Omega(R\mathbf{k})$, the total Berry curvature integrated over the BZ will be zero, resulting in a zero AHC. In our system, the mirror operation is a symmetry. We also performed DFT calculations to calculate the AHC for our system. The calculated AHC is close to zero. The magnetic point group is $-6'2'm$. We have also verified that the σ_{xy} component of the AHC tensor in this magnetic point group is zero using the MTENSOR module at Bilbao Crystallographic Server.

Revision: We have added a section in the SI to show this formula, symmetry analysis, and the DFT calculation of AHC for our system, along with a new figure (S3) that shows the magnetic structure and symmetry operations, and several corresponding references for the new work that we have carried out. We reproduce this section below:

4 Anomalous Hall Conductivity

The intrinsic anomalous Hall conductivity (AHC) can be calculated as:^{1,2}

$$\sigma_{xy} = -\frac{e^2}{\hbar} \int_{\text{BZ}} \frac{d\mathbf{k}}{(2\pi)^3} \Omega_{xy}(\mathbf{k}) \quad (\text{S1})$$

where $\Omega_{xy}(\mathbf{k})$ is the total Berry curvature

$$\Omega_{xy}(\mathbf{k}) = \sum_n f_n(\mathbf{k}) \Omega_{n,xy}(\mathbf{k}). \quad (\text{S2})$$

n is the band index, and $f_n(\mathbf{k})$ is the occupation. $\Omega_{n,xy}(\mathbf{k})$ is the Berry curvature for the n th band which is calculated as

$$\Omega_{n,xy}(\mathbf{k}) = -2\hbar^2 \text{Im} \sum_{m \neq n} \frac{\langle \psi_{n\mathbf{k}} | \hat{v}_x | \psi_{m\mathbf{k}} \rangle \langle \psi_{m\mathbf{k}} | \hat{v}_y | \psi_{n\mathbf{k}} \rangle}{[E_m(\mathbf{k}) - E_n(\mathbf{k})]^2}, \quad (\text{S3})$$

where \hat{v} is the velocity operator, ψ is the eigenstate, and E is energy. From the above equations, we see that AHC is determined by the total Berry curvature integrated over the Brillouin zone (BZ).

In magnetic materials, the Berry curvature and hence the intrinsic AHC is closely related to magnetic symmetry operations. Berry curvature behaves like an axial vector. If we apply a time-reversal operation, since the Berry curvature is odd with respect to a time-reversal operation $T\Omega(\mathbf{k}) = -\Omega(-\mathbf{k})$, the Berry curvature will change the sign. Applying a time-reversal operation is equivalent to reversing the direction of magnetization of each sublattice. If the magnetic system has a symmetry such that $R\Omega(\mathbf{k}) = -\Omega(\mathbf{k}')$, the Berry curvature is zero when integrating over the BZ, where R can be a mirror or rotation operation.³ Such symmetry analysis has been applied to noncollinear antiferromagnet GaNMn₃ and explicitly verified by DFT calculations as reported in Ref. [4].

Figure S3: Magnetic ordering in Cr_{1/4}TaS₂ determined by neutron scattering. (a) top view (b) side view. Mirror operations M_x and M_z are labeled as dashed lines. Red arrows denote the magnetization of Cr atoms.

Magnetic ordering in Cr_{1/4}TaS₂ is determined by neutron scattering and shown in Figure S3. The associated magnetic space group is $P\bar{6}'2'm$, of which we consider two magnetic symmetry operations, M_x and TM_z . Note that magnetization is also an axial vector, when applying a mirror operation, the component parallel to the mirror plane will reverse the direction, while the component perpendicular to the mirror plane will not. When applying these two symmetry operations to the Berry curvature, we have

$$\mathcal{M}_x\Omega^z(k_x, k_y, k_z) = -\Omega^z(-k_x, k_y, k_z) \quad (\text{S4})$$

$$\mathcal{T}\mathcal{M}_z\Omega^z(k_x, k_y, k_z) = \mathcal{T}\Omega^z(k_x, k_y, -k_z) = -\Omega^z(-k_x, -k_y, k_z). \quad (\text{S5})$$

Either symmetry operation makes the Berry curvature an odd function in the k space, resulting in a zero Berry curvature when integrating the BZ. Our calculated AHC from DFT calculations is 1.6 S/cm, which is close to zero. We have also verified that the σ_{xy} component of AHC tensor in the magnetic space group $P\bar{6}'2'm$ is zero using the MTENSOR module at the Bilbao Crystallographic Server.⁵

The AHC is calculated using the OpenMX code^{6,7} which utilizes pseudo-atomic orbitals as the basis functions.⁸ Cr6.0-*s3p2d1*, Ta7.0-*s3p2d2f1* and S7.0-*s2p2d1f1* are specified as the basis functions. A cutoff energy of 300 Ry and a k -point grid of $4 \times 4 \times 4$ are used for self-consistent calculation with spin-orbit coupling. The AHC is calculated using a $24 \times 24 \times 24$ k -point grid.

We have called out these equations and the corresponding figure in a new sentence towards the end of the “Synthesis and Crystallographic and Magnetic Structure” section:

More details of the symmetry analysis and the calculated AHC are presented in the Supplementary Information (Equations S1–S5 and Figure S3).

We also added References 1 through 8 in the Supplementary Information, cited in the section reproduced above:

1. Wang, X., Yates, J. R., Souza, I. & Vanderbilt, D. Ab initio calculation of the anomalous Hall conductivity by Wannier interpolation. *Phys. Rev. B* **74**, 195118 (2006).
2. Nagaosa, N., Sinova, J., Onoda, S., MacDonald, A. H. & Ong, N. P. Anomalous Hall effect. *Rev. Mod. Phys.* **82**, 1539–1592 (2010).
3. Suzuki, M.-T., Koretsune, T., Ochi, M. & Arita, R. Cluster multipole theory for anomalous Hall effect in antiferromagnets. *Phys. Rev. B* **95**, 094406 (2017).
4. Gurung, G., Shao, D.-F., Paudel, T. R. & Tsymbal, E. Y. Anomalous Hall conductivity of noncollinear magnetic antiperovskites. *Phys. Rev. Mater.* **3**, 044409 (2019).
5. Gallego, S. V., Etxebarria, J., Elcoro, L., Tasci, E. S. & Perez-Mato, J. M. Automatic calculation of symmetry-adapted tensors in magnetic and non-magnetic materials: a new tool of the Bilbao Crystallographic Server. *Acta Cryst. A* **75**, 438–447 (2019).
6. T. Ozaki et al., OpenMX 3.9. <http://www.openmx-square.org>.

7. Sawahata, H., Yamaguchi, N., Minami, S. & Ishii, F. First-principles calculation of anomalous Hall and Nernst conductivity by local Berry phase. *Phys. Rev. B* **107**, 024404 (2023).
8. Ozaki, T. Variationally optimized atomic orbitals for large-scale electronic structures. *Phys. Rev. B* **67**, 155108 (2003).

A more general comment: both data and experimental techniques are so many, in the paper, that at a certain point it results difficult to follow the logic behind the work. I would suggest to split the Results section in some sub-sections with titles addressing the type and/or technique of characterization.

Response: We thank the reviewer for their helpful suggestion to improve the organization of the manuscript. We agree that it would be organizationally beneficial to split up the Results section into subsections to make the paper easier to follow for readers.

Revision: We have separated the Results section into the following five subsections:

- Synthesis and crystallographic and magnetic structure
- Magnetotransport
- Local superlattice ordering
- Magnetometry
- Electronic structure

Reviewer #2 (Remarks to the Author):

The paper by Lilia S. Xie et al. addressed timely topic of different phases and their impact on the properties of the system in the TM-M-dichalcogenide group. These materials have been studied in the 70's and 80's (e.g. B. van Laar et al., S.T.T. Parkin et al.) and now experience a renaissance with the discovery of van der Waals materials and their engineering and topological aspects of the band structure. One of such interesting materials is Cr_{1/3}TaS₂, in which formation of chiral/heli magnetic ordering with phase transition at about 140K-150K was observed (doi.org/10.1063/5.0166333, [doi:10.1088/1742-6596/746/1/012061](https://doi.org/10.1088/1742-6596/746/1/012061), doi.org/10.1002/aelm.202100424). The Authors of this submission create yet a new phase: Cr_xTaS₂ composed of these elements, with x=1/4 and a new Cr sub-lattice (2 x 2) clearly resolved in the electron diffraction studies, incorporated in the known Cr_{1/3}TaS₂ phase with a pattern ($\sqrt{3} \times \sqrt{3}$). They evidence an intriguing anomalous Hall effect, that changes sign with temperature, which is a main discovery of the paper. Numerous methods are used to fully characterize the system, from calorimetric studies, neutron, X ray and electron diffraction, Raman scattering, magnetization and ARPES.

While I find the results interesting, timely and original, some parts of the paper are confusing and need reformulation and/or more in-depth discussion. Let me put my remarks in points, in a chronological order.

1. Abstract states “Chemical composition is commonly considered to be the primary determinant of superlattice identity, especially in intercalation compounds. Here, we find that, contrary to this conventional wisdom, kinetic control of superlattice growth leads to the coexistence of disparate domains within a compositionally “perfect” single crystal”. I find this message misleading. I do not see the points where Authors put that common wisdom in question. Instead, they rely on it: it is the correct chemical content that is fundamental for this research, it is a new phase, thermodynamically possible, which was demonstrated. Instead of contrast it would be more appropriate to speak about “an added value”.

Response: We thank the reviewer for this thoughtful feedback. Indeed, our findings are not incompatible with the notion that composition is an important determinant of superlattice identity.

Change: We have edited the sentence the reviewer pointed out to the following:

Nevertheless, in this work, we find that kinetic control of superlattice growth leads to the coexistence of disparate crystallographic domains within a compositionally “perfect” single crystal.

2. The claim of “microscopic patterning” in the context of self-organized sample with disorder is misleading. That would be authorized if the Authors present a series of samples with different cooling rates to tune proportions between two phases observed and demonstrate control of this new phase formation. Or there is any patterning that I missed here? Perhaps sample post-growth processing?

Response: We thank the reviewer for bringing up this point. We have not rigorously studied the effect of different cooling rates in this study, but previous work from our group (reference 31, Goodge, B. H., Gonzalez, O. et al. *ACS Nano* **17**, 19865–19876 (2023)) has shown that cooling rates can be used to tune defect distributions and magnetism in $\text{Cr}_{1/3}\text{NbS}_2$ and $\text{Cr}_{1/3}\text{TaS}_2$, two materials that are closely related to the compound studied in this manuscript, $\text{Cr}_{1/4}\text{TaS}_2$. Our group has also undertaken a computational study of the superlattice phase diagram (reference 63, Craig et al. arXiv:2410.19664v1) that further suggests cooling rates should allow control over phase distributions. Nevertheless, we agree with the reviewer’s point that we have not demonstrated deliberate patterning in this experimental work.

Revision: We have edited the sentence the reviewer pointed out to the following:

These results provide a blueprint for the deliberate engineering of macroscopic transport responses via microscopic tuning of magnetic exchange interactions in superlattice domains.

3. Referring to the electrical transport data, in $r_{xx}(T)$ there is a kink (it is not a “sharp drop”) evidencing magnetic phase transition. It is ascribed to transition to the antiferromagnetic state. What are the arguments supporting that it is in fact an antiferromagnetic phase in $\text{Cr}_{1/4}\text{TaS}_2$ that develops below 145 K ? Was there any magnetic anisotropy observed related to this phase? While the sample contains two phases, perhaps this transition occurs in $\text{Cr}_{1/3}\text{TaS}_2$ phase, especially since literature data evidence phase transition at 140-150 K in this material. Do the Authors have an idea why the second phase transition is not visible in the $r_{xx}(T)$ data? Also, why should this high temperature phase be absent below $\sim 100\text{K}$ and not contribute to the AHE? A comment should be provided.

Response: We are grateful to the reviewer for this thoughtful and detailed feedback. We primarily ascribe the change in $\rho_{xx}(T)$ data observed at 145 K to a bulk AFM transition on the basis of the neutron diffraction experiments, which unambiguously show the emergence of new Bragg peaks below 150 K. The spins were found to align within the ab plane on the basis of this neutron diffraction data as well: the in-plane orientation of the Cr moments yielded a stable refinement.

Although the transition to a chiral helimagnetic phase in $\text{Cr}_{1/3}\text{TaS}_2$ has indeed been demonstrated to occur around 145 K in the literature, multiple pieces of data from our experiments indicate that the transition we see at this temperature cannot be ascribed to $\text{Cr}_{1/3}\text{TaS}_2$. Firstly, X-ray, neutron, and electron diffraction experiments (Figures 1 and 3) all indicate that the majority of the sample consists of a 2×2 superlattice, which has a different symmetry from the $\sqrt{3} \times \sqrt{3}$ superlattice in $\text{Cr}_{1/3}\text{TaS}_2$. The heat capacity data (Figure 1c) shows that the transition observed at 145 K in our samples of $\text{Cr}_{1/4}\text{TaS}_2$ is a bulk transition, as we see a well-defined anomaly at this temperature. Thus, we attribute the transition to a magnetic transition associated with the 2×2 superlattice, and, by inference, not to the chiral helimagnetic transition in $\text{Cr}_{1/3}\text{TaS}_2$ with a different superlattice. For a similar reason, we believe the 100 K transition is not discernable in our $\rho_{xx}(T)$ data because the $\sqrt{3} \times \sqrt{3}$ domains responsible for this transition comprise a small minority of the sample by volume.

Regarding “why should this high temperature phase be absent below $\sim 100\text{K}$ and not contribute to the AHE,” if we understand the question correctly, the majority AFM phase indeed is present below 100 K, but we argue that it does not contribute to AHE on the basis of symmetry analysis of the magnetic structure. We discuss this in more detail in the Supplementary Information,

especially in the newly added section on the anomalous Hall conductivity calculation and symmetry analysis.

Revision: We have changed the description of the change in the $\rho_{xx}(T)$ data at 145 K to the following:

A kink at $T_N = 145$ K is consistent with reduced carrier scattering upon bulk AFM ordering.

We have also added a new subsection to the Supplementary Information, “4. Anomalous Hall Conductivity,” detailing the symmetry analysis of the magnetic structure and new calculations. We have included this new material in our answer to Reviewer #1 above, and will not duplicate it here again in the interest of concision.

As for the ferromagnetic phase, present below 100K, it is ascribed to the Cr_{1/3}TaS₂ phase with a pattern ($\sqrt{3} \times \sqrt{3}$) and magnetization in the c plane (Fig.3h). This phase is described as responsible for the appearance of the AHE signal. Why would the in-plane magnetization lead to the AHE (when B is applied perpendicular to plane) in spite of the in-plane magnetization? The evolution of M vs B-out-of-plane is shown in Fig.S7 and here the coercive field is much lower than this observed in the AHE, putting some doubts to the interpretation.

Response: We thank the reviewer for raising this point. We agree that the coercive field of the AHE signal should be compared to the coercive field of the magnetization with the field applied out-of-plane (i.e. in the same orientation). In the M vs. $\mu_0 H$ data with field applied out-of-plane, a small hysteresis/coercive on the order of 1 T is observed at 2 K, decreasing at increasing temperatures to less than 0.5 T at 80 K. The hysteresis is absent at 120 K. The size of the hysteresis at 80 K is of the same magnitude as the hysteresis observed in the AHE data, which is consistent with the AHE being linked to the presence of the ferromagnetic phase below 100 K. Differences in coercive fields observed in the M vs. $\mu_0 H$ data and the AHE data at other temperatures indicate that scattering between domains (and not just the magnetization of the ferromagnetic domains alone) is responsible for the AHE behavior.

Revision: We have added Figure S9 to the Supplementary Information, showing the isothermal M vs. $\mu_0 H$ data measured at 2 K, 40 K, 80 K, and 120 K. We show this figure and its caption below:

Figure S9: Isothermal magnetization curves with $\mu_0 H \parallel c$.

We have also added the following sentence to the Results section towards the end of the discussion of the magnetometry data:

The coercive field observed in the isothermal magnetization with $\mu_0 H \parallel c$ data is larger, as would be expected for the field applied along the hard axis (Figure S9), but comparable in magnitude to the AHE hysteresis between 50 and 100 K.

4. Authors study “in-plane Hall resistivity ρ_{xy} ”. Typically, it is the ρ_{yx} component of the resistivity tensor that is called Hall resistivity [e.g. C. L. Chien, C. R. Westgate, Hall Effect and its Applications]. Also, what does “in-plane” refers to? Is magnetic field applied in the plane of the layer? Basing on the information provided, it seems that magnetic field is oriented along c axis and it is the usual Hall effect.

Response: We thank the reviewer for raising these clarifying points and supplying this reference. For our transverse (Hall) resistivity measurements, we agree that ρ_{yx} is consistent with convention. “In-plane” refers to the orientation of the measurement leads in the ab plane of the crystal, and does not refer to the so-called planar Hall effect. The magnetic field is oriented along the c axis, as in the usual or conventional Hall effect geometry. We are grateful to the reviewer for pointing out this potentially misleading wording.

Revision: Throughout the manuscript, we have replaced our previous usage of ρ_{xy} to refer to the transverse resistivity with ρ_{yx} , as the reviewer suggested. We have also revised all usage of “in-plane Hall resistivity” and “in-plane longitudinal resistivity” to “Hall resistivity” and “longitudinal resistivity,” respectively, to reduce the potential for confusion.

5. Referring to the interpretation of the Hall resistivity: when anomalous Hall signal is present, it is often impossible to decompose the ordinary and anomalous components and to extract the values of concentration (the anomalous one happens to be present even at high field). This is the case here: the kink in the $n(T)$ dependence does not reflect change in concentration, but comes from the AHE contribution.

Response: We thank the reviewer for pointing out this possibility. We agree that it is difficult to deconvolute AHE contribution from changes in carrier concentration, and we cannot assign the changes in the Hall resistivity definitively based on the data.

Revision: We have edited a sentence in the Discussion section to point out this possibility:

No clear low-field signature of AHE is observed below 40 K, reminiscent of the disappearance of some components of the AHE in the cluster glass phase of the kagome 120° AFM Mn_3Sn (though we note that the kink in the apparent n_h observed below 50 K could suggest higher-field AHE contributions).

6. Discussing Anomalous Hall Effect the Authors rule out an intrinsic effect, related to the Berry curvature, as an origin of the AHE signal $T=145$ K, indicating that this component is zero. I think that this fact should be justified in a more elaborate way or references should be given to support this claim. The mirror symmetry is present only when the magnetic moments are flipped (so when time reversal symmetry is broken at the same time). It should be commented. Also, it is important to notice that S-O effects also contribute to the Berry curvature. Perhaps presented calculations may be extended to demonstrate vanishing Berry curvature integrals (at $B=0$)?

Response: We thank the reviewer for these comments. We agree that more extensive investigation of the intrinsic anomalous Hall conductivity would improve the paper.

Revision: As mentioned in an earlier response above, we have added a new subsection to the Supplementary Information, “4. Anomalous Hall Conductivity,” which includes more details and references for our symmetry analysis and the results of our calculated anomalous Hall conductivity, which is consistent with the symmetry analysis indicating vanishing Berry curvature. Spin-orbit coupling was included in the calculations. We have included this new material in our answer to Reviewer #1 above, and will not duplicate it here again in the interest of concision.

7. Referring to the AHE conductivity, it is an extra component of the conductivity: conductivity $s_{yx} = s_{\text{AHE}} + s_{yx}(\text{ordinary})$. The Hall conductivity transforms as $\sigma_{yx} = \rho_{yx} / (\rho_{xx}^2 + \rho_{yx}^2)$, where ρ_{yx} is experimentally found. My doubts concern the way the AHE conductivity

was calculated (replacing simply ρ_{xy} by ρ_{AHE}). In my opinion it is the total sigma that should be calculated and then anomalous component should be identified.

Response: We agree with the reviewer that the total Hall conductivity should be calculated using this formula. In the case of our material, where $\rho_{yx} \ll \rho_{xx}$ and $\rho_{AHE} \ll \rho_{yx}$, the values obtained are very similar, and the qualitative conclusions drawn from our analysis of the anomalous Hall conductivity remain unchanged. Nevertheless, we have revised our derivation of σ_{AHE} to involve first a calculation of the total σ_{xy} , and then subtraction of the ordinary Hall component.

Revision: We have revised the information about calculation of σ_{AHE} in the Results section to the following:

We calculate the anomalous Hall conductivity, σ_{AHE} , by first calculating the total transverse conductivity, $\sigma_{xy} = \frac{\rho_{yx}}{\rho_{yx}^2 + \rho_{xx}^2}$, and then subtracting the ordinary Hall component. The maximum σ_{AHE} is about $8 \Omega^{-1} \text{ cm}^{-1}$ at 50 K.

We have also updated Figure S14 in the Supplementary Information with the corrected values for σ_{AHE} , shown below:

8. I find the names: bulk and minority domains (e.g. in the abstract) and the opposition between them, confusing. It should be clearly stated whether 2×2 and $(\sqrt{3} \times \sqrt{3})$ domains are surface domains of bulk ones. It is not clear in which parts the Authors study few layer exfoliated flakes and where bulk samples.

Response: We thank the reviewer for pointing out the possible confusion arising from our choice of descriptors. We believe all domains studied in this work are bulk domains, not surface domains (the exfoliated sample studied in the 4D-STEM experiments is thick enough to rule out the exclusive contribution of surface domains). As indicated in the Methods section, only the 4D-STEM experiments were carried out on an exfoliated sample.

Revision: We have edited the sentence in question in the abstract to the following:

We demonstrate that $\text{Cr}_{1/4}\text{TaS}_2$ is a bulk noncollinear antiferromagnet in which scattering between majority and minority superlattice domains engenders complex magnetotransport below the Néel temperature, including an anomalous Hall effect.

9. I do not quite understand the role of the control sample, with $x=0.23$ and Cr deficiency. It has lower crystal quality, higher concentration, but still two phases with different Cr arrangements (and some Cr missing) evidenced in the Raman scattering. One would expect the AHE to be present, when it is absent for both. Was sample geometry the same of transport experiments? Why “higher carrier higher carrier concentration is consistent with the absence of FM domains” – that needs more justification. The possible way I see is though Berry phase dependence of Fermi level position, but Authors rule out this mechanism.

Response: We are grateful to the reviewer for raising these thoughtful points. We agree that the connection between carrier concentration and FM domains is not clear. Instead of domain structure per se, it is more reasonable to expect a different filling level (which can be attributed to a different stoichiometry in the Cr-deficient sample) changes the carrier concentration. The sample geometry was identical for both. Regarding the Raman data, a full assignment of peaks would require polarization-dependent measurements and a calculation of phonon mode energies that is beyond the scope of this study. However, we interpret the lack of a well-defined, sharp peak at 148 cm^{-1} (which is observed in the main $\text{Cr}_{1/4}\text{TaS}_2$ samples) as an indication that $\sqrt{3} \times \sqrt{3}$ domains are not conclusively present in the Cr-deficient sample.

Revision: We have revised the specific sentence the reviewer pointed out to the following:

The higher carrier concentration is consistent with larger hole pockets due to the lower Cr stoichiometry. We also observe lower mobility and lower MR, which are additionally consistent with lower crystal quality and the absence of FM/SG domains.

10. In Fig.3 f the curves for $H \parallel c$ are not visible at all – why not use right vertical scale?

Response: We thank the reviewer for this suggestion. Our objective in Figure 3f in showing the two different field orientations was to demonstrate that easy-plane behavior is observed (i.e. the

susceptibility values obtained with $\mu_0 H \parallel c$ are much lower). Thus, we opted to present them on the same vertical axis on this plot. We believe it would be challenging to use the right vertical axis here because we are also comparing the field-cooled vs. zero-field-cooled behavior in this panel, which would overlap significantly with the $\mu_0 H \parallel c$ data if we were to change the y -axis scale. However, we agree that it would be valuable to present the $\mu_0 H \parallel c$ susceptibility data in a more legible format.

Revision: We have added the χ_{DC} vs. T measured with $\mu_0 H \parallel c$ data to the Supplementary Information in Figure S3. The figure and caption are shown below:

Figure S3: Field-cooled (FC) and zero-field-cooled (ZFC) DC magnetic susceptibility (χ_{DC}) vs. T , measured with a 100 Oe field perpendicular to c .

We call out this figure in the Results section of the main text (“Magnetometry” subsection) in the following sentence:

The larger χ_{DC} with the magnetic field applied in-plane is consistent with the expected easy-plane behavior (Figure S3).

11. Instead of “Zero field hysteresis of less than 1T” one should say that coercive field is about 1T.

Response: We are grateful to the reviewer for suggesting this more precise term.

Revision: We have edited sentences in the Results section which previously contained the term “zero-field hysteresis” to the following:

| Between 40 to 100 K, ρ_{AHE} is positive in sign and exhibits a small coercive field of less than 1 T up to 80 K.

| At 4 K, M with $\mu_0 H \perp c$ shows a small coercive field of 400 Oe (Figure 3h inset).

We also have edited the caption of Figure S7 to the following:

| Isothermal magnetization curves with $\mu_0 H \perp c$, showing a non-zero coercive field at 94 K and below.

12. In Fig.S4 the frequency evolution of the peak at higher temperatures should be shown.

Response: A larger temperature range for the AC susceptibility data is shown in Figure 3g of the main text. We believe this illustrates the frequency evolution of the peak shown in Figure S4 at higher temperatures (up to 105 K). Our intention in showing a narrower temperature range in Figure S4 is to compare the fitting of the data (which we performed to extract the maximum of the trace for each frequency) with the AC susceptibility data itself.

13. Comment in the SI in the crystal growth section (see the Supplementary Information for more details on crystal growth and compositional characterization) is missing. Also, the description of the studied samples is not sufficient, are they bulk or exfoliated samples? For transport experiment, how were the hallbars prepared, what were the dimensions ?

Response: We thank the reviewer for drawing our attention to this error regarding the details on crystal growth and compositional characterization. We included this information in the Methods section, not the Supplementary Information. Regarding the Hall bars, information about preparation is included in the Methods section as well (“For electrical transport measurements, cleaved single crystals with thicknesses of 30 μm or less were affixed using GE Varnish and contacted using silver paint and gold wire.”). We agree that it would be helpful to add more information about dimensions.

Revision: We have edited the sentence that the reviewer pointed out about the crystal growth methods to the following:

| see the Methods for more details on crystal growth and compositional characterization.

We have also added a sentence later in the Methods section giving more details about typical dimensions for a Hall bar device:

| Typical dimensions were on the order of $50 \times 100 \times 20 \mu\text{m}$.

In addition to the Tables (s1 and s2) showing the presence of the a given phase, the XRD spectra should be presented in the supplementary. While in log scale, they evidence the lack of parasitic phases which could contribute to both magnetic and transport data

Overall, I think that the data and physics of this material is very interesting. However, being interesting it is also truly complex and more precise explanations are often needed for the reader to comprehend and appreciate the content. Therefore I would recommend the Authors to address the points I raised.

Response: We thank the reviewer for this suggestion. We have presented the precession images from the single-crystal X-ray diffraction data in Figure S1. Even in log scale, we did not observe reflections from any parasitic phases (either other superlattices or any other unexplained peaks). In our experience, it is common in intercalation compounds that weaker-intensity features are more readily observed in electron diffraction than X-ray diffraction under typical collection conditions for the latter (see, for example, Erodici et al., *J. Phys. Chem. C* **127**, 9787–9795 (2023)). For the Cr_xTaS_2 compounds, the host lattice reflections are much more intense than the intercalant superlattice reflections (which can be rationalized by the much higher Z value of Ta compared to Cr), so we believe it is unsurprising that typical exposure times on a laboratory diffractometer (that do not saturate the host lattice peaks) do not show reliably detectable reflections for minority superlattice phases. However, the 4D-STEM data (Figure 3c–e) unequivocally show that $\sqrt{3} \times \sqrt{3}$ domains are present and do indeed contribute the magnetic and transport behavior, as discussed throughout the manuscript.

We deeply appreciate the reviewer's positive appraisal of the data and physics presented in this work and their constructive suggestions to improve and further clarify the interpretations and explanations.

Reviewer #3 (Remarks to the Author):

In this work, the authors discuss the growth and characterization of Cr intercalated TaS₂ (Cr_{1/4}TaS₂). They identify a magnetic transition at ~145 K via a weak heat capacity signature and the magnetic structure via neutron diffraction as a 120 degree antiferromagnetic (AFM) phase. The very unusual properties of the material manifest in the presence of an anomalous Hall effect (AHE), common in ferromagnetic (FM) systems, but generally less common in antiferromagnetic ones owing to the lack of overall time reversal symmetry breaking. They

further characterize slight differences in the microscopic structures and ordering of the Cr intercalants that via Raman and transmission electron microscopy. The rest of the article constructs a toy model scenario around this Cr intercalated microstructure about coexisting AFM/FM domains that could potentially give rise to an AHE and the unusual magneto transport. As the anomalous Hall effect is a fascinating phenomenon that elucidates the relation between the topology of Bloch wavefunctions and their transport/scattering processes, the work in this manuscript could likely be important in this regard. The experimental work in growth and characterization is extensive, and the authors' dedication shows in the figures and text of the manuscript. There is perhaps a lack of detailed quantitative theory/justification of how the processes give rise to the AHE and magneto transport, especially the sign change of the AHE conductivity with temperature and something conclusive of how it scales with longitudinal conductivity, but perhaps this work indeed will spark future studies to address this question.

I recommend the article for publication once the authors address some detailed concerns as listed below.

1. Upon my initial reading of the abstract, I was confused about what “domain” meant in this context, as the title leads me to believe that AFM domain exist and that domain wall between them are responsible for this anomalous Hall effect. I initially thought it was referring to ordered magnetic domains (ie AFM), but was actually in reference to the Cr intercalants forming a 2×2 and $\sqrt{3} \times \sqrt{3}$ superlattice, which of course plays a key role in the different magnetic phases discussed later in the paper. The abstract should be clear about the type of domains studied - consider mentioning that the Cr intercalants form such 2×2 and $\sqrt{3} \times \sqrt{3}$ superlattice and mention how this is determined (via transmission electron microscopy). In addition, the abstract is a bit lean on what experimental techniques are involved (I count heat capacity, magnetotransport, Hall effect, Raman, neutron scattering, and magnetization) of which the authors should take pride in and mention a few.

Response: We are grateful to the reviewer for these thoughtful appraisals and bringing points of possible confusion to our attention. We agree that these clarifications and additional details would improve the quality and usefulness of the abstract.

Revision: We have edited the relevant sentences in the abstract to the following:

Nevertheless, in this work, we find that kinetic control of superlattice growth leads to the coexistence of disparate crystallographic domains within a compositionally “perfect” single crystal. We demonstrate that $\text{Cr}_{1/4}\text{TaS}_2$ is a bulk noncollinear antiferromagnet in which scattering between majority and minority superlattice domains engenders complex magnetotransport below the Néel temperature, including an anomalous Hall effect. We characterize the magnetic phases in different domains, image their nanoscale

morphology, and propose a mechanism for nucleation and growth using neutron diffraction, magnetotransport, transmission electron microscopy, and angle-resolved photoemission spectroscopy, in addition to density functional theory calculations and symmetry analysis.

2. On page 5 there is a sentence that reads “The in-plane Hall resistivity (ρ_{xy}) is dominated by the ordinary Hall effect” – this needs to be reworded. Although it is clear from the context of this paper that voltage, current and field are all perpendicular to each other, the concept of “in-plane” Hall effect is a real phenomenon that can manifest with the field in the xy plane. See for example: J. Zhou et al., Nature 609, 46-51 (2022) or even your own reference 66 Mayoh et al.

Response: We are grateful to the reviewer for pointing out this inaccurate wording. We agree that we should revise our terminology such that there is no possibility of misinterpreting our data as the planar Hall effect.

Revision: We have revised all usage of “in-plane Hall resistivity” and “in-plane longitudinal resistivity” to “Hall resistivity” and “longitudinal resistivity,” respectively, to reduce the potential for confusion.

3. On the MR of Fig. 2f, there appears to be hysteresis in some of the MR data (ie 120 K data presented in inset), although perhaps not believable for the 50 K data. However, it is not clear which of the traces correspond to the upswing and the downswing. Also, the 120 K MR data in Fig. 2f appears to show hysteretic behavior persisting up to +/- 2 T, but the 120 K AHE in Fig. 2d exhibits non hysteretic behavior (expected since FM onsets at 98 K) and saturates at a modest field of perhaps +/- 0.5 T. Could the authors comment on this discrepancy and compare/contrast saturation fields between MR and AHE, as well as presence/absence of hysteretic behavior between MR and AHE.

4. Related to the above point, is it clear where the current is flowing as the temperature is varied and the nature of the $\sqrt{3} \times \sqrt{3}$ minority magnetic state changes from PM to FM to spin glass (SG)? Presumably, the disappearance of AHE at low temperatures could be attributed not to the lack of AHE in the SG phase, but perhaps simply because it is much more resistive than the 2×2 AFM majority regions and we no longer obtain boundary scattering as the majority of the current just flows around the SG regions through the network of 2×2 AFM regions. The large positive (quadratic-like?) MR at 10 K relative to the negative MR at higher temperatures seems consistent with this picture as SG typically have negative MR (seen in your higher temperature data). Also, a minor issue: clarify if the antisymmetrization and symmetrization of AHE and MR, was done by mixing upsweeps and downsweeps? ie $\rho_{xy_up} (+H) \rightarrow 0.5 * (\rho_{xy_up} (+H) - \rho_{xy_down} (-H))$ and $\rho_{xx_down} (+H) \rightarrow 0.5 * (\rho_{xx_down} (+H) + \rho_{xx_up} (-H))$. I believe this is the only way hysteretic behavior can appear in your presented

AHE and MR data in Fig. 2, but please mention this explicitly in the methods section towards the end of the paper.

Response: We thank the reviewer for their attention to the detail of the MR data. Upon closer examination, the apparent hysteresis in the MR data was an artifact due to inconsistencies in the way we interpolated the data for forward and reverse sweeps. We have re-symmetrized the data shown in Figure 2f (with the protocols that the reviewer indicated), and do not observe hysteresis at either 50 K or 120 K (or, for that matter, at any temperature) beyond the noise level of our data. Thus, it does not appear that the hysteretic behavior observed in the AHE is reflected in the MR data. We cannot rule out the possibility that the lack of detectable hysteresis in the MR data may be a result of insufficient instrumental sensitivity.

We are also grateful for the reviewer’s insightful suggestion that the disappearance of AHE at low temperatures could be a result of more resistive behavior in the $\sqrt{3} \times \sqrt{3}$ superlattice regions. Without a pristine sample with the equivalent composition and this local structure, it is difficult to confirm, but this would indeed be consistent with our AHE and MR observations.

Revision: The updated MR data is provided in Figure 2f, with arrows indicating the sweep directions (and different colors/opacities for the forward/reverse sweeps):

We have also updated the Methods section to explicitly include the equations used for antisymmetrization and symmetrization of the ρ_{yx} and ρ_{xx} data, respectively:

ρ_{xx} and ρ_{yx} data were symmetrized and antisymmetrized, respectively, according to the following equations (with \rightarrow indicating the forward sweep direction, and \leftarrow indicating the negative sweep direction):

$$\rho_{yx,\text{symm}}^{\rightarrow}(\mu_0 H) = \frac{1}{2} [\rho_{yx}^{\rightarrow}(\mu_0 H) - \rho_{yx}^{\leftarrow}(-\mu_0 H)]$$

$$\rho_{yx,\text{symm}}^{\leftarrow}(\mu_0 H) = \frac{1}{2} [\rho_{yx}^{\leftarrow}(\mu_0 H) - \rho_{yx}^{\rightarrow}(-\mu_0 H)]$$

$$\rho_{xx,\text{symm}}^{\rightarrow}(\mu_0 H) = \frac{1}{2} [\rho_{xx}^{\rightarrow}(\mu_0 H) + \rho_{xx}^{\leftarrow}(-\mu_0 H)]$$

$$\rho_{xx,\text{symm}}^{\leftarrow}(\mu_0 H) = \frac{1}{2} [\rho_{xx}^{\leftarrow}(\mu_0 H) + \rho_{xx}^{\rightarrow}(-\mu_0 H)]$$

Magnetoresistance (MR) data were calculated according to the following equation:

$$\text{MR}(\mu_0 H) (\%) = \left[\frac{\rho_{xx}(\mu_0 H)}{\rho_{xx}(0)} - 1 \right] \times 100\%$$

We have also added the following sentence to the Discussion section, to propose the possibility that the different superlattice domains may have different temperature evolution of resistivity, which may contribute to the AHE and MR observations:

Finally, at low temperatures, the positive, quadratic-like MR and the disappearance of the low-field AHE could indicate that the $\sqrt{3} \times \sqrt{3}$ -containing domains become significantly more resistive than the 2×2 majority domains, resulting in the measured transport behavior corresponding to only the AFM 2×2 majority domains.

5. The sentence at the end of page 5 "The saturation magnitude of ρ_{AHE} increases with increasing T in this regime" is confusing, because one typically assumes that ρ_{AHE} is the zero-field extrapolated finite component and so the term "saturation" may be misleading. Also, given this definition, the vertical axis of Fig. 2c (ρ_{AHE}) is also misleading. The authors should just clarify the definitions and delineate extracted quantities from fit parameters.

Response: We are grateful to the reviewer for pointing out this potential source of confusion. Indeed, our usage of the term "saturation" in referring to the anomalous Hall resistivity is not

meant to correspond to a zero-field component, so we agree that a clearer term (and explicit definition) would be desirable.

Revision: We have edited the sentence that the reviewer pointed out to the following:

Above $\mu_0H > 1$ T, the ρ_{AHE} values are approximately constant; we plot the average values at each temperature in Figure 2d. At 100 K and below, these average high-field values of ρ_{AHE} increase with increasing T .

We have also correspondingly edited the y -axis of Figure 2d and the caption, which we reproduce below:

(d) The average high-field values of ρ_{AHE} ($\mu_0H > 1$ T) vs. T .

6. Regarding Figure 5 and the growth/high-T picture, although exact identification of the 2×2 and $\sqrt{3} \times \sqrt{3}$ formation temperatures would likely involve differential scanning calorimetry and/or thermogravimetric techniques, I believe faith in the authors' expertise is warranted and they can be forgiven for speculating the temperature ranges at which these metallurgical processes occur. This would allow them to replace/clarify the vague "High T" with something like $\sim 700 \text{ C} < T < 800 \text{ C}$, and similar for Intermediate T and Low T. Changes to the main text are also warranted in order to avoid the rather abrupt transition from previous discussion and crystal growth regime. Also take care in relabeling Fig. 5a temperature regimes so as to avoid confusing the reader with the previous "low T" data, as well as the pics in panel b), where one may think "high T" in panel a) corresponds with 98-145K in panel b).

Response: We appreciate the reviewer's thoughtful suggestions to improve the specificity and usefulness of the temperature ranges in which we believe the nucleation and growth processes of these phases within a crystal monolith may occur. We agree that the "high," "intermediate," and "low" temperature labels are somewhat unsatisfactory; however, we are unsure if we can

pinpoint the actual temperature ranges precisely enough to be meaningful. For example, in reference 32 (Lawrence et al., *Inorg. Chem.* **62**, 18179–18188 (2023)), a similar intercalation compound, $\text{Fe}_{1/4}\text{NbS}_2$, was annealed at 400 °C to induce intercalant superlattice disorder. In an older paper, Boswell et al., *Phys. Status Solidi A* **45**, 469–481 (1978), the authors observe the disappearance of 2×2 superlattice spots and the appearance of diffuse scattering at about 325 °C (also for $\text{Fe}_{1/4}\text{NbS}_2$). In a recent pre-print from our group, Craig et al., arXiv:2410.19664v1 (2024)), the authors find computationally that a disordered arrangement of intercalants is favored over the 2×2 superlattice above approximately 225 °C (albeit for closed-shell bilayer systems).

Given this spread in temperature ranges observed in the literature, along with the possibility that Cr^{3+} as an intercalant may have different diffusion characteristics compared to Fe^{2+} , we do not feel confident in being able to accurately specify ranges for “high,” “intermediate,” and “low” temperatures as they relate to superlattice formation. However, we agree that confusion about the distinction between these “metallurgical” temperatures and the temperatures given for the magnetic behavior would be important to avoid if possible.

Revision: In Figure 5, we have specified that “Low T ” refers to a range that is greater than 300 K, with the implication that “High T ” and “Intermediate T ” are also considerably higher temperatures compared to the temperatures at which we observe any magnetic ordering in this material. The new figure is shown below.

We have also added the Boswell et al. paper discussed above and the Craig et al. pre-print as new references (62 and 63), and cited them in the following sentence:

This picture explains how $\sqrt{3} \times \sqrt{3}$ order can be present in samples with compositions that should favor a 2×2 superlattice from purely thermodynamic considerations,^{32,62} consistent with computational predictions.⁶³

62. Boswell, F. W., Prodan, A., Vaughan, W. R. & Corbett, J. M. On the ordering of Fe atoms in Fe_xNbS_2 . *Phys. Status Solidi A* **45**, 469–481 (1978).

63. Craig, I. M., Kim, B. J., Limmer, D. T., Bediako, D. K. & Griffin, S. M. Modeling the superlattice phase diagram of transition metal intercalation in bilayer $2H\text{-TaS}_2$. arXiv.org arXiv:2410.19664v1 (2024).

7. Finally, regarding the discussion paragraph on page 13 and 14 talking about the AHE, I think this should be expanded on, as it is the key result of this work. The sign change of the AHE is quite interesting and points to something drastic happening with a change in transport through the $\sqrt{3} \times \sqrt{3}$ regions of the material at the PM/FM transition. As the $\sigma_{\text{AHE}}^{\text{xy}}$ scales linearly with σ_{xx} (Fig. S11) this system is perhaps sensitive to skew-scattering processes, not surprising given the magnetic scattering presumably present at this domain boundary. Its interesting that the magnitude of the apparent slope of $\sigma_{\text{AHE}}^{\text{xy}}$ with σ_{xx} is about the same before and above the PM to FM transition (ie the sign change), suggesting that the actual microscopic nature of the scattering process does not change nor the band structure/density of states (determined by the 2×2 band structure). Perhaps this is a coincidence, but perhaps a believable postulation can be drawn about a rather unusual change in orientation of the spins responsible for the skew scattering must change direction. Are the authors aware of other superlattice to superlattice boundaries of magnetic phases and local magnetic structures/defects that could possibly be present at this domain boundary? Any other examples of sign changes of the AHE in similar or dissimilar systems should also be cited.

Response: We thank the reviewer for their interest in this aspect of our work, namely the AHE sign change. The most representative work in the literature in which a similar sign change in AHE has been observed in a compound with very similar structure and chemistry is reference 16, Checkelsky, J. G., Lee, M., Morosan, E., Cava, R. J. & Ong, N. P. Anomalous Hall effect and magnetoresistance in the layered ferromagnet $\text{Fe}_{1/4}\text{TaS}_2$: The inelastic regime. *Phys. Rev. B* **77**, 014433 (2008). In addition, another work on $\text{Cr}_{1/3}\text{NbS}_2$ has also found a sign change in AHE, albeit at temperatures significantly lower than the ordering temperature of the material—this is reference 67, Bornstein, A. C., Chapman, B. J., Ghimire, N. J., Mandrus, D. G., Parker, D. S. & Lee, M. Out-of-plane spin-orientation dependent magnetotransport properties in the anisotropic helimagnet $\text{Cr}_{1/3}\text{NbS}_2$. *Phys. Rev. B* **91**, 184401 (2015). We have cited both of these papers in the Discussion section where we propose superlattice domain boundaries as a possible explanation of the AHE and MR behavior.

To our understanding, sign changes in AHE are a relatively general phenomenon that can be observed in numerous ferromagnetic metals, as well as noncollinear spin systems in which scalar spin chirality is often invoked. The former is discussed in some detail in reference 16, Checkelsky et al., and the latter is covered in the well-known review that is reference 64 in our manuscript, Nagaosa, N., Sinova, J., Onoda, S., MacDonald, A. H. & Ong, N. P. Anomalous Hall effect. *Rev. Mod. Phys.* **82**, 1539–1592 (2010). Nevertheless, given the observation of qualitatively similar sign changes in AHE in different intercalation compounds, we agree with the reviewer that this is worth emphasizing further.

Revision: We have added the following sentence to the Discussion section:

In general, the qualitative similarity of sign changes in the AHE of several intercalated TMDs suggests that superlattice boundaries may be a general source of skew scattering in these systems, and that further study of superlattice boundary effects on their magnetotransport properties is merited.^{16,67}

Reviewer #4 (Remarks to the Author):

We thank the reviewer for contributing their feedback and comments, which we hope to have addressed above.

Additional Revisions:

Over the course of addressing the reviewers' comments, we made some additional changes to the manuscript for accuracy and readability. We detail these below by section and subsection where relevant:

Introduction:

For clarity, we altered the construction of a sentence describing the tunability of magnetic phases in intercalated transition metal dichalcogenides. The new sentence is below:

In this family of materials, varying the host lattice, intercalant, and stoichiometry allows access to many types of magnetism, including hard ferromagnetism,^{15–17} noncollinear antiferromagnetism,^{18–20} and spin glass phases.^{21,22}

Results, Synthesis and crystallographic and magnetic structure; Author contributions:

We removed the unnecessary acronym “EDS” for “energy dispersive X-ray spectroscopy.”

Results, Magnetometry:

We discovered that we had mistakenly assigned the T_C of the ferromagnetic minority phase from the Arrott plot in Figure 3i as 98 K, instead of 96 K, which is the correct T_C . This has been corrected in Figure 3i and elsewhere in the manuscript, including the main and supplementary text, caption to Figure 3, and Figure 5b.

Results, Electronic structure:

In the caption of Figure 4, we used the acronym “ARPES” for “angle-resolved photoemission spectroscopy” without defining it. We have amended this. The new caption for Figure 4b is below:

(b) Experimental Fermi surfaces from angle-resolved photoemission spectroscopy (ARPES) measured with linear horizontal (LH) and linear vertical (LV) polarization.

Materials and Methods:

We have added previously missing details for the synthesis and preparation of samples of $\text{Cr}_{1/3}\text{TaS}_2$ and $2H\text{-TaS}_2$:

Single crystals of $\text{Cr}_{1/3}\text{TaS}_2$ were grown in an analogous fashion using a 0.47:1:2.1 molar ratio.

Single crystals of $2H\text{-TaS}_2$ were obtained from HQ Graphene and mechanically exfoliated onto SiO_2/Si .

Data Availability:

In accordance with the policy, we have added a Data Availability statement:

Crystallographic data for the structure reported in this article have been deposited at the Cambridge Structural Database under the deposition number 2388468. All other relevant data are included within the article and its supplementary information. Additional raw data files are available from the corresponding authors upon reasonable request.

Author Contributions:

For accuracy, we have edited the contributions for the collection and analysis of Raman spectroscopy data:

| L.S.X., M.P.E., C.M., and S.H. collected and analyzed Raman data.

In closing, we are grateful to all the reviewers for their thoughtful and constructive comments, which we believe we have addressed in this letter. We hope that our manuscript may now be suitable for publication in *Nature Communications*.

Please accept our sincere thanks for your time and effort in handling our manuscript.

Dear Dr. Bladwell,

Thank you for your consideration of our manuscript “Anomalous Hall effect from inter-superlattice scattering in a noncollinear antiferromagnet” (manuscript ID NCOMMS-24-70569-T) for publication in *Nature Communications*. We respond to the reviewers’ comments below.

Reviewer #1 (Remarks to the Author):

For what concerns the points I raised, authors made an excellent work to clarify them (in particular the AHC conductivity and its explanation by Berry curvature), improving the scientific clarity and content of the paper, that to my opinion can be published as is.

Reviewer #3 (Remarks to the Author):

The authors have done a wonderful job of addressing all concerns. I recommend proceeding to publication.

Reviewer #4 (Remarks to the Author):

We thank all the reviewers for their positive appraisals and also for their time and effort in reviewing our manuscript and providing constructive feedback that has improved our work.

Please accept our sincere thanks for your time and effort in handling our manuscript.